# COMPOUND RETURNS REDUCE VARIANCE IN REINFORCEMENT LEARNING

## ABSTRACT

Multistep returns such as $n$-step returns are commonly used to improve the sample efficiency of deep reinforcement learning (RL). Variance becomes the limiting factor in the length of the returns; looking too far into the future increases uncertainty and reverses the benefit of multistep learning. In our work, we study the ability of compound returns—weighted averages of $n$-step returns—to reduce variance. The $\lambda$-return, used by TD($\lambda$), is the most well-known compound return. We prove for the first time that any compound return with the same contraction rate as a given $n$-step return has strictly lower variance when experiences are not perfectly correlated. Because the $\lambda$-return is expensive to implement in deep RL, we also introduce an approximation called Piecewise $\lambda$-Return (PiLaR), formed by averaging two $n$-step returns, that offers similar variance reduction while being efficient to implement with minibatched experience replay. We conduct experiments showing PiLaRs can train Deep Q-Networks faster than $n$-step returns with little additional computational cost.

## 1 INTRODUCTION

Efficiently learning value functions is critical for reinforcement learning (RL) algorithms. Value-based RL methods (e.g., Watkins, 1989; Rummery & Niranjan, 1994; Mnih et al., 2015) encode policies implicitly in a value function, and therefore policy evaluation is the principal mechanism of learning. Even when RL methods instead learn parametric policies, accurate value functions are needed to guide policy improvement (e.g., Silver et al., 2014; Lillicrap et al., 2016; Fujimoto et al., 2018; Haarnoja et al., 2018) or to serve as a baseline (e.g., Barto et al., 1983; Sutton, 1984; Williams, 1992; Schulman et al., 2015a). Quicker policy evaluation is therefore useful to many RL algorithms.

One way to achieve faster value-function learning is through multistep returns, in which more than one reward following an action is used for reinforcement. Multistep returns have been extensively used in deep RL (e.g. Schulman et al., 2015b; Mnih et al., 2016; Munos et al., 2016; Schulman et al., 2017; Hessel et al., 2018; Schrittwieser et al., 2020; Chebotar et al., 2023; Schwarzer et al., 2023), where the value function is approximated by a neural network. In theory, multistep returns incorporate additional information regarding future outcomes, leading to faster credit assignment and, in turn, faster learning. However, faster learning is not guaranteed in practice because looking farther into the future increases uncertainty and can end up requiring more samples. These two opposing factors must be balanced to achieve fast and stable learning. The most common multistep returns are $n$-step returns and $\lambda$-returns, both of which span between standard temporal-difference (TD; Sutton, 1988) and Monte Carlo (MC) learning through the choice of value of $n$ or $\lambda$.

The main consideration in choosing which of these return estimators to use has generally been implementation. In classic RL, the value function is a lookup table or a linear parametric function, so the $\lambda$-return is preferred for its efficient implementation using TD($\lambda$) with eligibility traces (Sutton, 1988). However, in deep RL, the value function is a neural network trained with an experience replay buffer (Lin, 1992), so the extra bootstrapping performed by the $\lambda$-return becomes costly and $n$-step returns are more common. Although recent work has explored ways to efficiently train neural networks using replayed $\lambda$-returns (e.g., Munos et al., 2016; Harb & Precup, 2016; Daley & Amato, 2019), $\lambda$-returns generally remain more expensive or complex to implement than $n$-step returns.

Despite its challenging implementation in deep RL, the $\lambda$-return is interesting because it averages many $n$-step returns together. This averaging might reduce variance and therefore lead to faster

learning than $n$-step returns, but this has never been shown. More generally, the $\lambda$-return is an example of a *compound* return, or a weighted average of two or more $n$-step returns (Sutton & Barto, 2018, sec. 12.1). While compound returns are understood to be theoretically sound, they have not been extensively analyzed. These averages are ideally suited for deep RL because experiences are already stored in a replay memory, making it easy to compute several $n$-step returns cheaply and then average them together in any desired way.

In this paper, we formally investigate the variance of compound returns. We derive the first variance model for arbitrary compound returns. Our assumptions are a significant relaxation of previous work that modeled $n$-step variance (Konidaris et al., 2011). We further prove that all compound returns have a *variance-reduction property*; any compound return that achieves the same contraction rate as an $n$-step return has strictly lower variance if the TD errors are not perfectly correlated, which is almost always the case in practice. As a corollary, there exists a $\lambda$-return for every $n$-step return that achieves the same contraction rate but lower variance, implying a better bias-variance trade-off.

Because the $\lambda$-return remains expensive to implement for deep RL, we propose an efficient approximation called Piecewise $\lambda$-Returns (PiLaRs). A PiLaR is computed by averaging just two $n$-step returns together—the most efficient compound return possible. The lengths of the $n$-step returns are chosen to put weights on the TD errors that are close to those assigned by TD($\lambda$), thereby achieving similar variance reduction as the $\lambda$-return. We show that PiLaRs improve sample efficiency compared to $n$-step returns when used to train Deep Q-Network (DQN; Mnih et al., 2015), and we expect to see similar improvements for other value-based deep RL methods.

## 2 BACKGROUND

Value-based RL agents interact with their environments to iteratively improve estimates of their expected cumulative reward. By convention, the agent-environment interaction is modeled as a Markov decision process (MDP) described by the tuple $(\mathcal{S}, \mathcal{A}, P, R)$. At each time step $t$ of the process, the agent observes the environment state, $S_t \in \mathcal{S}$, and selects an action, $A_t \in \mathcal{A}$, accordingly. The environment then transitions to a new state, $S_{t+1} \in \mathcal{S}$, with probability $P(S_{t+1} \mid S_t, A_t)$, and returns a scalar reward, $R_t \stackrel{\text{def}}{=} R(S_t, A_t, S_{t+1})$. We assume the agent samples each action with probability $\pi(A_t|S_t)$, where $\pi$ is a stochastic policy.

In the standard prediction problem, the agent's goal is to learn a value function $V \colon \mathcal{S} \to \mathbb{R}$ that estimates the expected discounted return $v_\pi(s)$ attainable in each state $s$. Letting $G_t \stackrel{\text{def}}{=} \sum_{i=0}^{\infty} \gamma^i R_{t+i}$ be the observed Monte Carlo (MC) return at time $t$, where $\gamma \in [0, 1]$ is a discount factor and actions are implicitly sampled from $\pi$, then $v_\pi(s) \stackrel{\text{def}}{=} \mathbb{E}[G_t \mid S_t = s]$. The basic learning operation is a *backup*, which has the general form

$$V(S_t) \leftarrow V(S_t) + \alpha_t \left( \hat{G}_t - V(S_t) \right) , \tag{1}$$

where $\hat{G}_t$ is a return estimate (the *target*) and $\alpha_t \in (0, 1]$ is a step-size hyperparameter. Substituting various estimators for $\hat{G}_t$ leads to different learning properties. For instance, the MC return could be used, but it suffers from high variance, and delays learning until the end of an episode. To reduce the variance and delay, return estimates can *bootstrap* from the value function. Bootstrapping is the fundamental mechanism underlying TD learning (Sutton, 1988). The most basic multistep version of TD learning uses the $n$-step return as its target in Eq. (1):

$$G_t^n \stackrel{\text{def}}{=} R_t + \gamma R_{t+1} + \cdots + \gamma^{n-1} R_{t+n-1} + \gamma^n V(S_{t+n}). \tag{2}$$

Bootstrapping introduces significant bias in the update, since generally $\mathbb{E}\left[V(S_{t+n}) \mid S_t\right] \neq v_\pi(S_{t+n})$ due to estimation error, but it greatly reduces variance. The case of $n = 1$ corresponds to the classic TD(0) algorithm, $V(S_t) \leftarrow V(S_t) + \alpha_t \delta_t$, where $\delta_t \stackrel{\text{def}}{=} R_t + \gamma V(S_{t+1}) - V(S_t)$ is the TD error. However, bootstrapping after just one reward is slow because long-term reward information must propagate indirectly through the value function, requiring many behavioral repetitions before $V$ approximates $v_\pi$ well. Larger values of $n$ consider more rewards per update and assign credit faster, but at the price of increased variance, with $n \to \infty$ reverting to the MC return. The choice of $n$ thus faces a bias-variance trade-off (Kearns & Singh, 2000), with intermediate values typically performing best in practice (Sutton & Barto, 2018, ch. 7). Another type of multistep return is the $\lambda$-return, used by TD($\lambda$) algorithms (Sutton, 1988), which is equivalent to an exponentially

weighted average of $n$-step returns:

$$G_t^\lambda \overset{\text{def}}{=} (1 - \lambda) \sum_{n=1}^{\infty} \lambda^{n-1} G_t^n \,, \tag{3}$$

where $\lambda \in [0, 1]$. This form of the $\lambda$-return looks difficult to compute, but it has a recursive structure that enables efficient sequential computation (see Section 4). The $\lambda$-return is just one way to average $n$-step returns together, but any weighted average is possible. Such averages are known as *compound* returns (Sutton & Barto, 2018, ch. 12). Formally, a compound return is expressed as

$$G_t^{\mathbf{w}} \overset{\text{def}}{=} \sum_{n=1}^{\infty} w_n G_t^n \,, \tag{4}$$

where $\mathbf{w}$ is an infinite sequence of nonnegative weights with the constraint $\sum_{n=1}^{\infty} w_n = 1$. Eq. (4) is a strict generalization of all of the aforementioned return estimators; however, it technically constitutes a *compound* return if and only if at least two of the weights are nonzero—otherwise, it reduces to an $n$-step return. All choices of weights that sum to 1 are equally valid in the sense that using their compound returns as the target in Eq. (1) will converge to $v_\pi$ under the standard step-size criteria (Robbins & Monro, 1951). However, in practice, this choice greatly affects the bias and variance of the estimator and, as a consequence, the empirical rate of convergence to $v_\pi$. Furthermore, different choices of weights will vary in the amount of computation needed; sparse weights require less bootstrapping, and therefore less computation when the value function is expensive. A principal goal of this paper is to shed light on the trade-offs between these factors and to develop new compound returns that favorably balance all of these considerations (see Section 4).

Most deep RL agents are control agents, meaning they do not just predict $v_\pi$ for a fixed policy, but also use these estimates to update the behavior policy during training. One way to do this is Q-Learning (Watkins, 1989). Rather than learning a state-value function $V$, the agent learns an action-value function $Q \colon \mathcal{S} \times \mathcal{A} \to \mathbb{R}$ that estimates the expected return $q_*(s, a)$ earned by following an optimal policy after taking action $a$ in state $s$. The estimated value of a state is therefore $V(s) = \max_{a \in \mathcal{A}} Q(s, a)$, and all of the previously discussed return estimators apply after making this substitution. Backups are conducted as before, but now operate on $Q(S_t, A_t)$ instead of $V(S_t)$.

Learning is off-policy in this setting, since the agent now predicts returns for a greedy policy that differs from the agent's behavior. Any multistep return is therefore biased unless the expectation is explicitly corrected, e.g., by importance sampling (Kahn & Harris, 1951). However, it is common practice to ignore this bias in deep RL, and recent research has even suggested that doing so is more effective with both $n$-step returns (Hernandez-Garcia & Sutton, 2019) and $\lambda$-returns (Daley & Amato, 2019; Kozuno et al., 2021), the latter of which become Peng's Q($\lambda$) (Peng & Williams, 1994). For these reasons, we also forego off-policy corrections in our work.

## 3 VARIANCE ANALYSIS

Our main goal of this section is to derive conditions for when a compound return reduces variance compared to a given $n$-step return. We call this the *variance-reduction property* of compound returns (see Theorem 1). An important consequence of this property is that when these conditions are met and both returns are chosen to have the same contraction rate, the compound return needs fewer samples than the $n$-step return to converge (see Theorem 2).

Before we start, we note that our real quantity of interest in this section is the variance of the backup error $\hat{G}_t - V(S_t)$ conditioned on the state $S_t$. The degree of this quantity's deviation from its expected value is what ultimately impacts the performance of value-based RL methods in Eq. (1). Nevertheless, this turns out to be the same as the variance of the return $\hat{G}_t$, since $V(S_t)$ contributes no randomness when conditioned on the state $S_t$, and therefore $\text{Var}[\hat{G}_t - V(S_t) \mid S_t] = \text{Var}[\hat{G}_t \mid S_t]$. This equivalence allows us to interchange between the variances of a return and its error, depending on which is more computationally convenient. For brevity, we omit the condition on $S_t$ throughout our analysis, but it should be assumed for all variances and covariances.

Modeling the variance of a compound return is challenging because it typically requires making assumptions about how the variances of $n$-step returns increase as a function of $n$, as well as how

strongly correlated different $n$-step returns are. If these assumptions are too strong, the derived variances will fail to reflect reality and lead to poorly informed algorithmic choices. For instance, consider the following simple example of a compound return: $(1-w)G_t^1 + wG_t^2$, where $w \in (0, 1)$. Let $\sigma_n^2 \stackrel{\text{def}}{=} \text{Var}[G_t^n]$. The variance of this compound return is

$$(1-w)^2 \sigma_1^2 + w^2 \sigma_2^2 + 2w(1-w)\sigma_1 \sigma_2 \, \text{Corr}[G_t^1, G_t^2] \,.$$

To evaluate this expression, it would be tempting to assume either $\text{Corr}[G_t^1, G_t^2] = 0$ to remove the covariance term, or $\text{Corr}[G_t^1, G_t^2] = 1$ because both returns are generated from the same trajectory. However, neither assumption captures the underlying complexity of the average, since $(1-w)G_t^1 + wG_t^2 = (1-w)G_t^1 + w\left(G_t^1 + \gamma\delta_{t+1}\right) = G_t^1 + w\gamma\delta_{t+1}$. Thus, averaging these returns by choosing $w$ can mitigate only the variance due to $\delta_{t+1}$, but the randomness of $G_t^1$ can never be averaged out. Two different $n$-step returns are therefore neither uncorrelated nor perfectly correlated; they consist of various random elements, some of which are shared and cannot be averaged, and others which are not shared and can be averaged. Any model that fails to account for this partial averaging will lead to poor variance predictions. To be accurate, our variance assumptions must start from the most fundamental unit of randomness within a return: the TD error. We therefore propose the following uniformity assumptions on the TD errors.

**Assumption 1.** *TD errors have uniform variance:* $\text{Var}[\delta_i] = \kappa, \ \forall \, i$.

**Assumption 2.** *Pairs of distinct TD errors are uniformly correlated:* $\text{Corr}[\delta_i, \delta_j] = \rho, \ \forall \, i, \, j \neq i$.

These assumptions are reasonable, as they would be difficult to improve without invoking specific information about the MDP's transition function or reward function. We note that Konidaris et al. (2011) similarly made Assumption 1 for their $n$-step return variance model. However, in comparison, Assumption 2 is a significant relaxation and generalization of the other assumptions made by their model, which we show later in this section.

Note that Assumption 2 implies $\rho \geq 0$, because three or more TD errors cannot be negatively correlated simultaneously. Furthermore, the assumption is equivalent to $\text{Cov}[\delta_i, \delta_j] = \rho\kappa$ for all $i$ and $j \neq i$. Since $\text{Var}[\delta_i] = \text{Cov}[\delta_i, \delta_i]$, we can combine Assumptions 1 and 2 into one statement:

$$\text{Cov}[\delta_i, \delta_j] = ((1-\rho)\mathbf{1}_{i=j} + \rho)\kappa \,, \tag{5}$$

where $\mathbf{1}_{i=j}$ is 1 when $i = j$ and is 0 otherwise. In the proposition below, we derive a variance model for the $n$-step return by decomposing the return into a sum of discounted TD errors and then adding up the pairwise covariances given by Eq. (5). For brevity, we define a function $\Gamma_c(n)$ to represent the sum of the first $n$ terms of a geometric series with common ratio $\gamma^c$, where $c > 0$:

$$\Gamma_c(n) \stackrel{\text{def}}{=} \begin{cases} (1 - \gamma^{cn}) / (1 - \gamma^c), & \text{for } 0 \leq \gamma < 1 \,, \\ n, & \text{for } \gamma = 1 \,. \end{cases} \tag{6}$$

**Proposition 1** ($n$-step variance)**.** *Under Assumptions 1 and 2, the variance of an $n$-step return is*

$$\text{Var}\left[G_t^n\right] = (1 - \rho)\,\Gamma_2(n)\,\kappa + \rho\,\Gamma_1(n)^2\,\kappa \,.$$

We include all proofs in Appendix B. Our $n$-step variance model linearly interpolates between an optimistic case where TD errors are uncorrelated ($\rho = 0$) and a pessimistic case where TD errors are maximally correlated ($\rho = 1$). In the maximum-variance scenario of $\gamma = 1$, we have $\Gamma_1(n) = \Gamma_2(n) = n$, so the model becomes $(1 - \rho)n\kappa + \rho n^2\kappa$. We further note that the optimistic case with $\rho = 0$ recovers the $n$-step variance model from Konidaris et al. (2011, sec. 3), showing that the assumptions made in their model are equivalent to assuming that all TD errors are uncorrelated. In this case, the $n$-step variance increases (sub)linearly with $n$ when $\gamma$ is equal to (less than) 1.

Another advantage of our assumptions is that they allow us to go beyond $n$-step returns and calculate variances for arbitrary compound returns. We accomplish this by again decomposing the return into a weighted sum of TD errors (see the next lemma) and then applying Assumptions 1 and 2 to derive a compound variance model in the following proposition.

**Lemma 1.** *A compound error can be written as a weighted summation of TD errors:*

$$G_t^{\mathbf{w}} - V(S_t) = \sum_{i=0}^{\infty} \gamma^i W_i \delta_{t+i} \,, \quad \text{where } W_i \stackrel{\text{def}}{=} \sum_{n=i+1}^{\infty} w_n \,.$$

**Proposition 2** (Compound variance). *Under Assumptions 1 and 2, the variance of a compound return is*

$$\operatorname{Var}\left[G_t^{\mathbf{w}}\right] = (1 - \rho) \sum_{i=0}^{\infty} \gamma^{2i} W_i^2 \kappa + \rho \sum_{i=0}^{\infty} \sum_{j=0}^{\infty} \gamma^{i+j} W_i W_j \kappa \,. \tag{7}$$

The cumulative weights $\{W_i\}_{i=0}^{\infty}$ fully specify the variance of a compound return. For instance, the $\lambda$-return assigns a cumulative weight of $W_i = \sum_{n=i+1}^{\infty} (1-\lambda)\lambda^{n-1} = \lambda^i$ to the TD error $\delta_{t+i}$, which matches the TD($\lambda$) algorithm (Sutton, 1988). Substituting this weight into Eq. (7) and solving the geometric series yields the variance for the $\lambda$-return (see Appendix C).

## 3.1 THE VARIANCE-REDUCTION PROPERTY OF COMPOUND RETURNS

Proposition 2 provides a method of calculating variance, but we would still like to show that compound returns reduce variance relative to $n$-step returns. To do this, we first need a way to relate them in terms of their expected learning performance. This is because low variance by itself is not sufficient for fast learning; for example, the 1-step return has very low variance, but learns slowly.

In the discounted setting, a good candidate for learning speed is the *contraction rate* of the expected update. The contraction rate is a constant factor by which the maximum value-function error between $V$ and $v_\pi$ is guaranteed to be reduced. When the contraction rate is less than 1, the return estimator exhibits an *error-reduction property*: i.e., the maximum error decreases on average with every backup iteration of Eq. (1). This property is commonly used in conjunction with the Banach fixed-point theorem to prove that $V$ eventually converges to $v_\pi$ (see, e.g., Bertsekas & Tsitsiklis, 1996, sec. 4.3). The error-reduction property of $n$-step returns was first identified by Watkins (1989) and is expressed formally as

$$\max_{s \in \mathcal{S}} |\mathbb{E}\left[G_t^n \mid S_t = s\right] - V(s)| \leq \gamma^n \max_{s \in \mathcal{S}} |V(s) - v_\pi(s)| \,. \tag{8}$$

The contraction rate is the coefficient on the right hand side of the inequality—in this case, $\gamma^n$. Increasing $n$ reduces the contraction rate, increasing the expected convergence rate. The error-reduction property of $n$-step returns implies a similar property for compound returns, since a convex combination of $n$-step returns cannot have a higher contraction rate than that of its shortest $n$-step return. Although Sutton & Barto (2018, sec. 12.1) mentions that compound returns can be shown to have this error-reduction property, it has never been made explicit to our knowledge, and so we formalize it in the following proposition.

**Proposition 3** (Compound error-reduction property). *Any compound return satisfies the inequality*

$$\max_{s \in \mathcal{S}} |\mathbb{E}\left[G_t^{\mathbf{w}} \mid S_t = s\right] - V(s)| \leq \left(\sum_{n=1}^{\infty} w_n \gamma^n\right) \max_{s \in \mathcal{S}} |V(s) - v_\pi(s)| \,. \tag{9}$$

Proposition 3 shows the contraction rate of a compound return is the weighted average of the individual $n$-step returns' contraction rates. We can therefore find an $n$-step return that has the same error-reduction property as a given compound return by solving the equation

$$\gamma^n = \sum_{n'=1}^{\infty} w_{n'} \gamma^{n'} \tag{10}$$

for $n$. We call this the *effective $n$-step* of the compound return, since the compound return reduces the worst-case value-function error as though it were an $n$-step return whose length is the solution to Eq. (10). Note that the effective $n$-step assumes the weights $\mathbf{w}$ are chosen such that this solution is an integer—otherwise rounding is necessary and the contraction rates will not be exactly equal.

In undiscounted settings, we cannot directly equate contraction rates like this. When $\gamma = 1$, the contraction rate of any return becomes $\sum_{n'=1}^{\infty} w_{n'} = 1$. Fortunately, even though contraction rates cannot be directly compared in undiscounted problems, we can still solve the limit of Eq. (10) as $\gamma \to 1$ to define the effective $n$-step in this setting. We show this in the following proposition.

**Proposition 4** (Effective $n$-step of compound return). *Let $G_t^{\mathbf{w}}$ be any compound return and let*

$$n = \begin{cases} \log_\gamma \left(\sum_{n'=1}^{\infty} w_{n'} \gamma^{n'}\right), & \text{if } 0 < \gamma < 1 \,, \\ \sum_{n'=1}^{\infty} w_{n'} n', & \text{if } \gamma = 1 \,. \end{cases} \tag{11}$$

$G_t^{\mathbf{w}}$ *shares the same bound in Eq.* (9) *as the $n$-step return $G_t^n$ if $n$ is an integer.*

We refer to the quantity $\sum_{n'=1}^{\infty} w_{n'} n'$ as the *center of mass* (COM) of the return, since it is the first moment of the weight distribution over $n$-step returns. Intuitively, this represents the average length into the future considered by the return. With this definition, we are now ready to formalize the *variance-reduction property* of compound returns in the next theorem.

**Theorem 1** (Variance-reduction property of compound returns). *Consider a compound return $G_t^{\mathbf{w}}$ and an $n$-step return $G_t^n$ with $n$ defined by Eq.* (11). *Under Assumptions 1 and 2, the inequality* $\mathrm{Var}[G_t^{\mathbf{w}}] \leq \mathrm{Var}[G_t^n]$ *always holds. Furthermore, the inequality is strict whenever $\rho < 1$.*

Theorem 1 shows that whenever a compound return has the same contraction rate ($\gamma < 1$) or COM ($\gamma = 1$) as an $n$-step return, it has lower variance as long as the TD errors are not perfectly correlated. Perfect correlation between all TD errors would be unlikely to occur except for contrived, maximum-variance MDPs; thus, compound returns reduce variance in virtually all practical cases. Crucially, variance reduction is achieved for any type of weighted average—although the magnitude of reduction does depend on the specific choice of weights. The exact amount, in terms of $\kappa$, could be calculated by subtracting the compound variance from the $n$-step variance for a given contraction rate or COM. An interesting avenue for future work is to derive weights that maximize the variance reduction magnitude under Assumptions 1 and 2.

## 3.2  FINITE-TIME ANALYSIS

To prove that lower variance does lead to faster learning, we conduct a finite-time analysis of compound TD learning. We consider linear function approximation, where $V(s) = \phi(s)^{\top}\theta$ for features $\phi(s) \in \mathbb{R}^d$ and parameters $\theta \in \mathbb{R}^d$; tabular methods can be recovered using one-hot features. The parameters are iteratively updated according to

$$\theta_{t+1} = \theta_t + \alpha g_t^{\mathbf{w}}(\theta_t), \quad \text{where } g_t^{\mathbf{w}}(\theta) \stackrel{\text{def}}{=} \left(G_t^{\mathbf{w}} - \phi(S_t)^{\top}\theta\right)\phi(S_t) . \tag{12}$$

Our theorem generalizes recent analysis of 1-step TD learning (Bhandari et al., 2018, theorem 2).

**Theorem 2** (Finite-Time Analysis of Compound TD Learning). *Suppose TD learning with linear function approximation is applied under an i.i.d. state model (see Assumption 3 in Appendix B.2) using the compound return estimator $G_t^{\mathbf{w}}$ as its target. Let $\beta \in [0, 1)$ be the contraction rate of the estimator (see Proposition 3), and let $\sigma^2 \geq 0$ be the variance of the estimator under Assumptions 1 and 2. Assume that the features are normalized such that $\|\phi(s)\|_2^2 \leq 1$, $\forall\, s \in \mathcal{S}$. Define $C \stackrel{\text{def}}{=} (\|R\|_{\infty} + (1+\gamma)\|\theta^*\|_{\infty}) / (1 - \gamma)$, where $\theta^*$ is the minimizer of the projected Bellman error for $G_t^{\mathbf{w}}$. For any $T \geq (4 / (1 - \beta))^2$ and a constant step size $\alpha = 1 / \sqrt{T}$,*

$$\mathbb{E}\left[\left\|V_{\theta^*} - V_{\bar{\theta}_T}\right\|_D^2\right] \leq \frac{\|\theta^* - \theta_0\|_2^2 + 2(1-\beta)^2 C^2 + 2\sigma^2}{(1-\beta)\sqrt{T}} , \quad \text{where } \bar{\theta}_T \stackrel{\text{def}}{=} \frac{1}{T}\sum_{t=0}^{T-1}\theta_t .$$

With a constant step size, compound TD learning (and hence $n$-step TD learning as a special case) reduces the value-function error at the same asymptotic rate of $O(1/\sqrt{T})$ for any return estimator. However, both the contraction rate $\beta$ and the return variance $\sigma^2$ greatly influence the magnitude of the constant that multiplies this rate. Given an $n$-step return and a compound return with the same contraction rate, the compound return has lower variance by Theorem 1 and therefore converges faster to its respective TD fixed point.

## 3.3  CASE STUDY: $\lambda$-RETURNS

Although the $\lambda$-return is often associated with its efficient implementation using TD($\lambda$) and eligibility traces, our theory indicates that pursuing $\lambda$-returns for faster learning via variance reduction is also promising. We empirically support our theory by demonstrating the benefit of $\lambda$-returns in the random walk experiment from Sutton & Barto (2018, sec. 12.1). In this environment, the agent begins in the center of a linear chain of 19 connected states and can move either left or right. The agent receives a reward only if it reaches one of the far ends of the chain ($-1$ for the left, $+1$ for the right), in which case the episode terminates. The agent's policy randomly moves in either direction with equal probability. We train the agents for 10 episodes, updating the value functions after each episode with offline backups like Eq. (1). To pair the $n$-step returns and $\lambda$-returns together, we derive the effective $\lambda$ of an $n$-step return in the following proposition.

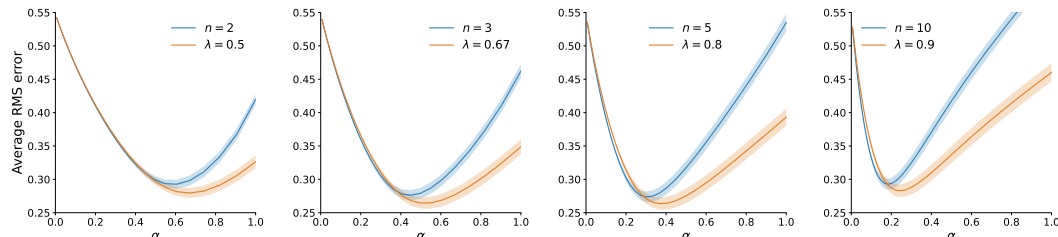

Figure 1: Comparing $n$-step returns and $\lambda$-returns, paired by equal COMs, in a random walk.

**Proposition 5** (Effective $\lambda$ of $n$-step return). *For any $n \geq 1$, the $\lambda$-return with $\lambda = (1 - \gamma^{n-1}) / (1 - \gamma^n)$ has the same contraction rate as the $n$-step return when $\gamma < 1$, and the $\lambda$-return with $\lambda = (n - 1) / n$ has the COM as the $n$-step return when $\gamma = 1$.*

Because this is an undiscounted task, we use the mapping $n \mapsto (n - 1) / n$ to enumerate several $(n, \lambda)$-pairs with equal COMs in Table 1. For our experiment, we choose four commonly used $n$-step values $\{2, 3, 5, 10\}$, which correspond to $\lambda \in \{0.5, 0.67, 0.8, 0.9\}$. In Figure 1, we plot the average root-mean square (RMS) value error (with respect to $v_\pi$) as a function of the step size $\alpha$ over 100 trials. We also indicate 95% confidence intervals by the shaded regions. For all of the tested $(n, \lambda)$-pairs, an interesting trend emerges. In the left half of each plot, variance is not an issue because the step size is small and has plenty of time to average out the randomness in the estimates. Learning therefore progresses at a nearly identical rate for both the $n$-step return and the $\lambda$-return since they have the same COM (although there is a small discrepancy as $\lambda \to 1$ due to the truncation of the episodic task, which reduces the $\lambda$-return's COM). However, as $\alpha \to 1$ in the right half of each plot, variance becomes a significant factor as the step size becomes too large to mitigate the noise in the updates. This causes the $n$-step return's error to diverge sharply compared to the $\lambda$-return's as the $\lambda$-return manages variance more effectively. Overall, the $\lambda$-return is never significantly worse than the

Table 1: $n$-step returns and $\lambda$-returns with equal COMs.

| $n$ | $\lambda$ |
|-----|-----------|
| 1 | 0 |
| 2 | 0.5 |
| 3 | 0.67 |
| 4 | 0.75 |
| 5 | 0.8 |
| 10 | 0.9 |
| 20 | 0.95 |
| 25 | 0.96 |
| 50 | 0.98 |
| 100 | 0.99 |

$n$-step return, and often much better. The best performance attained by the $\lambda$-return is also better than that of the $n$-step return in all cases. Notably, neither Assumption 1 nor 2 holds perfectly in this environment, demonstrating that our variance model's predictions are useful in practical settings.

## 4 EFFICIENTLY APPROXIMATING THE $\lambda$-RETURN

Although the previous experiment shows that $\lambda$-returns promote faster learning, they remain expensive for deep RL. This is because the $\lambda$-return at time $t$ theoretically bootstraps on every time step after $t$ (until the end of an episode), with each bootstrap requiring a forward pass through the neural network. While the $\lambda$-return can be truncated to reduce computation, it still requires $N$ bootstraps compared to just one for an $n$-step return if $N$ is the truncation length. Previous work has amortized the cost of $\lambda$-returns over long trajectories by exploiting their recursive structure (e.g., Munos et al., 2016; Harb & Precup, 2016; Daley & Amato, 2019). This works because $\delta_t^\lambda = \delta_t + \gamma \lambda \delta_{t+1}^\lambda$, where $\delta_t^\lambda \overset{\text{def}}{=} G_t^\lambda - V(S_t)$ is the TD($\lambda$) error, so the return at time $t$ can be computed using only one bootstrap given that $\delta_{t+1}^\lambda$ has already been computed. The price to pay for this efficiency is the requirement that experiences must be temporally adjacent, and it can be seen from the ablation experiment in Daley & Amato (2019, fig. 2) that the resulting sample correlations do hurt performance. Our preliminary experiments confirmed this, indicating that the correlations within replayed trajectories counteract the benefits of $\lambda$-returns when compared to minibatches of $n$-step returns (with the batch size chosen to equalize computation).

We instead seek a compound return that approximates the variance-reduction property of the $\lambda$-return while being computationally efficient for random-minibatch replay. There are many ways we could average $n$-step returns together, and so we constrain our search by considering compound returns that 1) comprise an average of only two $n$-step returns to minimize computational cost, 2) preserve the contraction rate or COM of the $\lambda$-return, and 3) place weights on the TD errors that are close to those assigned by the $\lambda$-return—i.e., TD($\lambda$).

The first property constrains our estimator to have the form $(1-w)G_t^{n_1} + wG_t^{n_2}$, where $1 \leq n_1 < n_2$ and $w \in [0,1]$. Let $n$ be our targeted effective $n$-step; the effective $\lambda$ can be obtained from Proposition 5. Let us further assume $\gamma < 1$, since deep RL methods commonly discount rewards (see Appendix D for the case where $\gamma = 1$). To preserve the contraction rate as in the second property, we must satisfy $(1-w)\gamma^{n_1} + w\gamma^{n_2} = \gamma^n$. Assuming that we have freedom in the choice of $n_1$ and $n_2$, it follows that $w = (\gamma^n - \gamma^{n_1}) / (\gamma^{n_2} - \gamma^{n_1})$. We would thus like to find $n_1$ and $n_2$ such that the weights given to the TD errors optimize some notion of closeness to the TD($\lambda$) weights, in order to fulfill the third and final property. Although there are many ways we could define the error, we propose to minimize the maximum absolute difference between the weights, since this ensures that no individual weight deviates too far from the TD($\lambda$) weight. Recall that the weight given to TD error $\delta_{t+i}$ by $n$-step return $G_t^n$ is $\gamma^i$ if $i < n$ and is 0 otherwise. It follows that our average assigns a weight of $\gamma^i$ if $i < n_1$ (since $(1-w)\gamma^i + w\gamma^i = \gamma^i$), $w\gamma^i$ if $n_1 \leq i < n_2$, and 0 otherwise. In comparison, TD($\lambda$) assigns a weight of $(\gamma\lambda)^i$. Our error function is therefore

$$\text{ERROR}(n_1, n_2) \stackrel{\text{def}}{=} \max_{i \geq 0} \left| W_i - (\gamma\lambda)^i \right|, \quad \text{where } W_i = \begin{cases} \gamma^i, & \text{if } i < n_1, \\ w\gamma^i, & \text{else if } i < n_2, \\ 0, & \text{else.} \end{cases} \quad (13)$$

We call our approximation Piecewise $\lambda$-Return (PiLaR) because each weight $W_i$ is a piecewise function whose value depends on where $i$ lies in relation to the interval $[n_1, n_2]$. Figure 2 illustrates how PiLaR roughly approximates the TD($\lambda$) decay using a step-like shape. Although PiLaR's TD-error weights do not form a smooth curve, they retain important properties like contraction rate, monotonicity, and variance reduction. Crucially, a PiLaR is significantly cheaper to compute than a $\lambda$-return, making it more suitable for minibatch experience replay. In Appendix D, we describe a basic search algorithm for finding the best $(n_1, n_2)$-pair according to Eq. (13), along with a reference table of precomputed PiLaRs for $\gamma = 0.99$.

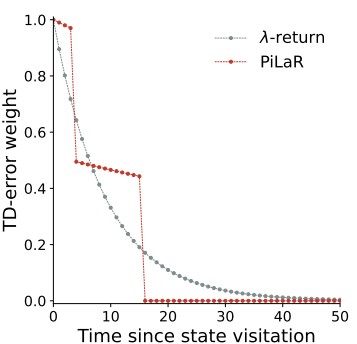

Figure 2: TD-error weights for a PiLaR and a $\lambda$-return ($\lambda = 0.904$). Both returns have the same contraction rate as a 10-step return when $\gamma = 0.99$.

## 5 DEEP RL EXPERIMENTS

We consider a multistep generalization of the Deep Q-Network (DQN) architecture (Mnih et al., 2015). The action-value function $Q(s, a; \theta)$ is implemented as a neural network to enable generalization over high-dimensional states, where $\theta \in \mathbb{R}^d$ is the learnable parameters. A stale copy $\theta^-$ of the parameters is used only for bootstrapping and is infrequently updated from $\theta$ in order to stabilize learning. The agent interacts with its environment and stores each MDP transition $(s, a, r, s')$—where $s$ is the state, $a$ is the action, $r$ is the reward, and $s'$ is the next state—in a replay memory $D$. The network's loss function is defined as

$$\mathcal{L}(\theta, \theta^-) \stackrel{\text{def}}{=} \mathbb{E}\left[ \frac{1}{2} \left( \hat{G}(r, s', \dots; \theta^-) - Q(s, a; \theta) \right)^2 \right],$$

where $\hat{G}(r, s', \dots; \theta^-)$ is a multistep return estimator for $q_*(s, a)$ and the expectation is taken over a uniform distribution on $D$. The network is trained by minimizing this loss using stochastic gradient descent (e.g., Adam; Kingma & Ba, 2015) on random minibatches of experiences sampled from $D$.

We test our agents in four MinAtar games (Young & Tian, 2019): Breakout, Freeway, Seaquest, and Space Invaders. The states are represented by $10 \times 10$ multi-channel images depicting object locations and velocity trails. The agents' network is a two-layer convolutional architecture with rectified linear units (ReLUs), and we add a dueling output layer (Wang et al., 2016) to improve action-value estimation. The agents execute an $\epsilon$-greedy policy for 5M time steps (where $\epsilon$ is linearly annealed from 1 to 0.1 over the first 100k steps) and conduct a minibatch update of 32 samples every 4 steps. We provide more details in Appendix E. Code is included in the supplementary material.

For $n \in \{3, 5, 10\}$, we compare the $n$-step return against the corresponding PiLaR of the same contraction rate for the given discount factor, $\gamma = 0.99$ (see Table 2 in Appendix D for specific values

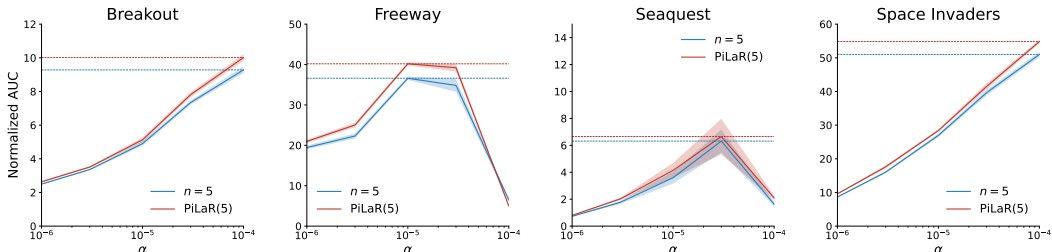

Figure 3: Step-size sensitivity curves for $n$-step returns and PiLaRs in four MinAtar games ($n = 5$).

of $n_1$, $n_2$, and $w$). We chose five Adam step sizes from a logarithmic grid search and generated learning curves by plotting the 100-episode moving average of undiscounted return versus time steps. We then calculated the area under each curve (AUC), normalized by training duration, and plotted these values against the step size ($\alpha$). We showcase these plots for $n = 5$ in Figure 3 and include all of the plots in Appendix E, including the learning curves. The results are averaged over 32 trials, with the shaded regions showing 95% confidence intervals. The horizontal dashed lines indicate the maximum performance achieved by each agent.

Although each pair of $n$-step return and PiLaR has the same contraction rate, PiLaRs are able to improve performance compared to $n$-step returns, supporting our hypothesis that a two-return average is able to appreciably reduce variance. The results somewhat mirror those from our earlier random walk experiment; when the step size is very small, there is no discernible difference between the estimators, but their performances begin to diverge as the step size grows and variance becomes more significant. In almost all cases (10 out of 12), PiLaR increases the agent's maximum performance with the best step size. Furthermore, the average performance gap across $\alpha$-values appears to widen as $n$ increases, suggesting that PiLaR's benefit becomes more pronounced for longer—and, hence, higher-variance—$n$-step returns. These results help to corroborate our theory and show that the variance reduction from PiLaRs can accelerate learning even for nontrivial network architectures.

## 6 CONCLUSION

We have shown for the first time that compound returns, including $\lambda$-returns, have a variance-reduction property. This is the first evidence, to our knowledge, that $\lambda$-returns have a theoretical learning advantage over $n$-step returns in the absence of function approximation; it was previously believed that both were different yet equivalent ways of interpolating between TD and MC learning. Our random walk experiments confirm that an appropriately chosen $\lambda$-return performs as least as well as—and often better than—a given $n$-step return across the entire range of step sizes. In replay-based deep RL methods like DQN, where $\lambda$-returns are difficult to implement efficiently, we demonstrated with PiLaR that a simpler average is still able to effectively reduce variance and train neural networks faster. Since the average is formed from only two $n$-step returns, the increase in computational cost is negligible compared to $n$-step DQN—less expensive than adding a second target network, as is often done in recent methods (e.g., Fujimoto et al., 2018; Haarnoja et al., 2018).

A number of interesting extensions to our work are possible. For instance, we derived PiLaR under the assumption that the $\lambda$-return is a good estimator to approximate. However, the exponential decay of the $\lambda$-return originally arose from the need for an efficient online update rule using eligibility traces, and is not necessarily optimal in terms of the bias-variance trade-off. With experience replay, we are free to average $n$-step returns in any way we want, even if the average would not be easily implemented online. This opens up exciting possibilities for new families of return estimators: e.g., those that minimize variance for a given contraction rate or COM. Based on our compound variance model (Proposition 2), a promising direction in this regard appears to be weights that initially decay faster than the exponential function but then slower afterwards. Minimizing variance becomes even more important for off-policy learning, where the inclusion of importance-sampling ratios greatly exacerbates variance. Recent works (Munos et al., 2016; Daley et al., 2023) have expressed arbitrary off-policy corrections in terms of weighted sums of TD errors, and so our theory could be extended to this setting with only minor modifications.

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

## A    HOW REALISTIC IS THE PROPOSED VARIANCE MODEL?

Assumptions 1 and 2 state that all TD errors have uniform variance and are equally correlated to one another. Since these assumptions may be violated in practice, it is informative to test how well our $n$-step variance model (Proposition 1) compares to the true $n$-step variances in several examples.

We consider three environments: the 19-state random walk (see Section 3), a $4\times3$ gridworld (Russell & Norvig, 2009, fig. 17.1), and a $10\times8$ gridworld (Sutton & Barto, 2018, fig. 7.4). We choose these environments because they have known dynamics and are small enough to allow exact calculation of $v_\pi$ with dynamic programming. The two gridworlds are stochastic because each of the four moves (up, down, left, right) succeeds with only probability 80%; otherwise, the move is rotated by 90 degrees in either direction with probability 10% each. We let the agent execute a uniform-random behavior policy for all environments.

To make the results agnostic to any particular learning algorithm, we use $v_\pi$ to compute the TD errors. We apply a discount factor of $\gamma = 0.99$ to the $10 \times 8$ gridworld (otherwise $v_\pi(s)$ would be constant for all $s$ due to the single nonzero reward) and leave the other two environments undiscounted. We then measure the variance of the $n$-step returns originating from the initial state of each environment, for $n \in [1, 21]$. Figure 4 shows these variances plotted as a function of $n$ and averaged over 10k episodes. The best-case (optimistic, $\rho = 0$) and worst-case (pessimistic, $\rho = 1$) variances predicted by the $n$-step model, assuming that $\kappa = \mathrm{Var}[\delta_0]$, are also indicated by dashed lines.

For all of the environments, the measured $n$-step variances always remain within the lower and upper bounds predicted by Proposition 1. These results show that our $n$-step variance model can still make useful variance predictions even when Assumptions 1 and 2 do not hold. The variances also grow roughly linearly as a function of $n$, corresponding more closely to the linear behavior of the optimistic, uncorrelated case than the quadratic behavior of the pessimistic, maximally correlated case. This further suggests that the majority of TD-error pairs are weakly correlated in practice, which makes sense because temporally distant pairs are unlikely to be strongly related.

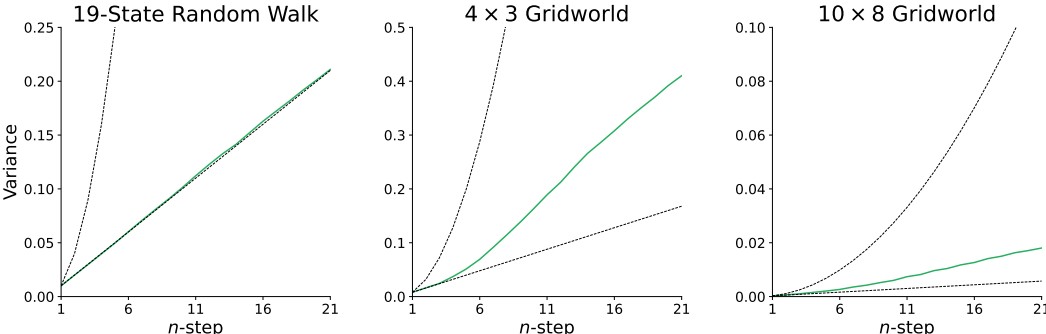

Figure 4:    Variances of the $n$-step returns originating from the initial state in three environments. The solid green line indicates the true variance while the dashed black lines indicate the lower and upper bounds predicted by our $n$-step variance model.

# B PROOFS

In this section, we include all proofs that were omitted from the main paper due to space constraints.

## B.1 VARIANCE MODELING

This section contains proofs related to return-variance modeling and the variance-reduction property.

### B.1.1 PROPOSITION 1

**Proposition 1** ($n$-step variance). *Under Assumptions 1 and 2, the variance of an $n$-step return is*

$$\mathrm{Var}\,[G_t^n] = (1 - \rho)\,\Gamma_2(n)\,\kappa + \rho\,\Gamma_1(n)^2\,\kappa\,.$$

*Proof.* The $n$-step error can be expressed as a finite summation of TD errors:

$$G_t^n - V(S_t) = \sum_{i=0}^{n-1} \gamma^i \delta_{t+i}\,. \tag{14}$$

Using this, we calculate the covariance between two $n$-step returns with lengths $n_1$, $n_2$:

$$
\begin{aligned}
\mathrm{Cov}\,[G_t^{n_1}, G_t^{n_2}] &= \mathrm{Cov}\left[\sum_{i=0}^{n_1-1} \gamma^i \delta_{t+i}, \sum_{j=0}^{n_2-1} \gamma^j \delta_{t+j}\right] \\
&= \sum_{i=0}^{n_1-1}\sum_{j=0}^{n_2-1} \mathrm{Cov}\left[\gamma^i \delta_{t+i}, \gamma^j \delta_{t+j}\right] \\
&= \sum_{i=0}^{n_1-1}\sum_{j=0}^{n_2-1} \gamma^{i+j} \mathrm{Cov}\left[\delta_{t+i}, \delta_{t+j}\right] \\
&= \sum_{i=0}^{n_1-1}\sum_{j=0}^{n_2-1} \gamma^{i+j}\left((1-\rho)\mathbf{1}_{i=j} + \rho\right)\kappa \\
&= (1-\rho)\sum_{i=0}^{\min(n_1,n_2)-1} \gamma^{2i}\kappa + \rho \sum_{i=0}^{n_1-1}\sum_{j=0}^{n_2-1} \gamma^{i+j}\kappa \\
&= (1-\rho)\sum_{i=0}^{\min(n_1,n_2)-1} \gamma^{2i}\kappa + \rho \sum_{i=0}^{n_1-1}\gamma^i \sum_{j=0}^{n_2-1} \gamma^j\kappa \\
&= (1-\rho)\,\Gamma_2(\min(n_1,n_2))\,\kappa + \rho\,\Gamma_1(n_1)\,\Gamma_1(n_2)\,\kappa\,. \tag{15}
\end{aligned}
$$

Because $\mathrm{Var}\,[G_t^n] = \mathrm{Cov}\,[G_t^n, G_t^n]$, then by letting $n_1 = n_2 = n$, we obtain the $n$-step variance formula and the proof is complete. $\square$

### B.1.2 LEMMA 1

**Lemma 1.** *A compound error can be written as a weighted summation of TD errors:*

$$G_t^{\mathbf{W}} - V(S_t) = \sum_{i=0}^{\infty} \gamma^i W_i \delta_{t+i}\,, \quad \textit{where } W_i \stackrel{def}{=} \sum_{n=i+1}^{\infty} w_n\,.$$

*Proof.* Let $W_i \stackrel{def}{=} \sum_{n=i+1}^{\infty} w_n$. We decompose the compound error into a weighted average of $n$-step errors, and then decompose those $n$-step errors into weighted sums of TD errors using Eq. (14):

$$G_t^{\mathbf{W}} - V(S_t) = \left(\sum_{n=1}^{\infty} w_n G_t^n\right) - V(S_t)$$

$$= \sum_{n=1}^{\infty} w_n \left( G_t^n - V(S_t) \right)$$

$$= \sum_{n=1}^{\infty} w_n \sum_{i=0}^{n-1} \gamma^i \delta_{t+i}$$

$$= w_1 \delta_t + w_2(\delta_t + \gamma \delta_{t+1}) + w_3(\delta_t + \gamma \delta_{t+1} + \gamma^2 \delta_{t+2}) + \dots$$

$$= (w_1 + w_2 + \dots) \delta_t + \gamma(w_2 + w_3 + \dots) \delta_{t+1} + \gamma^2(w_3 + w_4 + \dots) \delta_{t+2} + \dots$$

$$= W_0 \delta_t + \gamma W_1 \delta_{t+1} + \gamma^2 W_2 \delta_{t+2} + \dots$$

$$= \sum_{i=0}^{\infty} \gamma^i W_i \delta_{t+i},$$

which completes the lemma. $\qquad \square$

### B.1.3 PROPOSITION 2

**Proposition 2** (Compound variance). *Under Assumptions 1 and 2, the variance of a compound return is*

$$\mathrm{Var}\left[ G_t^{\mathbf{W}} \right] = (1 - \rho) \sum_{i=0}^{\infty} \gamma^{2i} W_i^2 \kappa + \rho \sum_{i=0}^{\infty} \sum_{j=0}^{\infty} \gamma^{i+j} W_i W_j \kappa. \qquad (7)$$

*Proof.* From Lemma 1, the variance of the compound return is

$$\mathrm{Var}\left[ G_t^{\mathbf{W}} \right] = \sum_{i=0}^{\infty} \sum_{j=0}^{\infty} \mathrm{Cov}\left[ \gamma^i W_i \delta_{t+i}, \gamma^j W_j \delta_{t+j} \right]$$

$$= \sum_{i=0}^{\infty} \sum_{j=0}^{\infty} \gamma^{i+j} W_i W_j \mathrm{Cov}\left[ \delta_{t+i}, \delta_{t+j} \right]$$

$$= \sum_{i=0}^{\infty} \sum_{j=0}^{\infty} \gamma^{i+j} W_i W_j ((1 - \rho)\mathbf{1}_{i=j} + \rho) \kappa$$

$$= (1 - \rho) \sum_{i=0}^{\infty} \gamma^{2i} W_i^2 \kappa + \rho \sum_{i=0}^{\infty} \sum_{j=0}^{\infty} \gamma^{i+j} W_i W_j \kappa,$$

which establishes Eq. (7). $\qquad \square$

### B.1.4 PROPOSITION 3

**Proposition 3** (Compound error-reduction property). *Any compound return satisfies the inequality*

$$\max_{s \in \mathcal{S}} |\mathbb{E}\left[ G_t^{\mathbf{W}} \mid S_t = s \right] - V(s)| \leq \left( \sum_{n=1}^{\infty} w_n \gamma^n \right) \max_{s \in \mathcal{S}} |V(s) - v_\pi(s)|. \qquad (9)$$

*Proof.* We expand the definition of a compound return in the left-hand side and then simplify:

$$\max_{s \in \mathcal{S}} \left| \mathbb{E}\left[ \sum_{n=1}^{\infty} w_n G_t^n \mid S_t = s \right] - V(s) \right| = \max_{s \in \mathcal{S}} \left| \mathbb{E}\left[ \sum_{n=1}^{\infty} w_n G_t^n \mid S_t = s \right] - \sum_{n=1}^{\infty} w_n V(s) \right|$$

$$= \max_{s \in \mathcal{S}} \left| \sum_{n=1}^{\infty} w_n \left( \mathbb{E}[G_t^n \mid S_t = s] - V(s) \right) \right|$$

$$\leq \sum_{n=1}^{\infty} w_n \max_s |\mathbb{E}[G_t^n \mid S_t = s] - V(s)|$$

$$\leq \left(\sum_{n=1}^{\infty} w_n \gamma^n\right) \max_{s \in \mathcal{S}} |V(s) - v_\pi(s)|\,,$$

where the first inequality is Jensen's inequality and the second inequality follows from the $n$-step error-reduction property of Eq. (8). This concludes the proof. $\qquad\square$

### B.1.5 PROPOSITION 4

**Proposition 4** (Effective $n$-step of compound return). *Let $G_t^{\mathbf{W}}$ be any compound return and let*

$$n = \begin{cases} \log_\gamma\left(\sum_{n'=1}^{\infty} w_{n'} \gamma^{n'}\right)\,, & \text{if } 0 < \gamma < 1\,, \\ \sum_{n'=1}^{\infty} w_{n'} n'\,, & \text{if } \gamma = 1\,. \end{cases} \tag{11}$$

*$G_t^{\mathbf{W}}$ shares the same bound in Eq. (9) as the $n$-step return $G_t^n$ if $n$ is an integer.*

*Proof.* When $\gamma < 1$, we can take the logarithm of both sides of Eq. (10) to get

$$n = \log_\gamma\left(\sum_{n=1}^{\infty} w_n \gamma^n\right) = \frac{\log\left(\sum_{n=1}^{\infty} w_n \gamma^n\right)}{\log \gamma}\,. \tag{16}$$

For the undiscounted case, we would like to evaluate this expression at $\gamma = 1$; however, since $\sum_{n=1}^{\infty} w_n = 1$ from the definition of a compound return, we arrive at an indeterminate form, $\frac{0}{0}$. Instead, we can apply L'Hôpital's rule to evaluate the limit as $\gamma \to 1$:

$$\lim_{\gamma \to 1} \frac{\log\left(\sum_{n=1}^{\infty} w_n \gamma^n\right)}{\log \gamma} = \lim_{\gamma \to 1} \frac{\frac{d}{d\gamma} \log\left(\sum_{n=1}^{\infty} w_n \gamma^n\right)}{\frac{d}{d\gamma} \log \gamma}$$

$$= \lim_{\gamma \to 1} \frac{\left(\sum_{n=1}^{\infty} w_n \gamma^{n-1} n\right) / \left(\sum_{n=1}^{\infty} w_n \gamma^n\right)}{1 / \gamma}$$

$$= \lim_{\gamma \to 1} \frac{\sum_{n=1}^{\infty} w_n \gamma^n n}{\sum_{n=1}^{\infty} w_n \gamma^n}$$

$$= \frac{\sum_{n=1}^{\infty} w_n n}{\sum_{n=1}^{\infty} w_n}$$

$$= \sum_{n=1}^{\infty} w_n n\,,$$

where the last step follows again from the fact that $\sum_{n=1}^{\infty} w_n = 1$. This establishes the case where $\gamma = 1$ and completes the proof. $\qquad\square$

### B.1.6 THEOREM 1

**Theorem 1** (Variance-reduction property of compound returns). *Consider a compound return $G_t^{\mathbf{W}}$ and an $n$-step return $G_t^n$ with $n$ defined by Eq. (11). Under Assumptions 1 and 2, the inequality $\mathrm{Var}[G_t^{\mathbf{W}}] \leq \mathrm{Var}[G_t^n]$ always holds. Furthermore, the inequality is strict whenever $\rho < 1$.*

*Proof.* Eq. (15) gives us an expression for the covariance between two $n$-step returns. We use this to derive an alternative formula for the variance of a compound return:

$$\mathrm{Var}\left[G_t^{\mathbf{W}}\right] = \sum_{i=1}^{\infty} \sum_{j=1}^{\infty} \mathrm{Cov}\left[w_i G_t^i, w_j G_t^j\right]$$

$$= \sum_{i=1}^{\infty} \sum_{j=1}^{\infty} w_i w_j \mathrm{Cov}\left[G_t^i, G_t^j\right]$$

$$= \sum_{i=1}^{\infty} \sum_{j=1}^{\infty} w_i w_j \big((1 - \rho)\, \Gamma_2(\min(i, j)) + \rho\, \Gamma_1(i)\, \Gamma_1(j)\big) \kappa$$

$$= (1 - \rho) \sum_{i=1}^{\infty} \sum_{j=1}^{\infty} w_i w_j \, \Gamma_2(\min(i,j)) \, \kappa + \rho \sum_{i=1}^{\infty} \sum_{j=1}^{\infty} w_i w_j \, \Gamma_1(i) \, \Gamma_1(j) \, \kappa \, .$$

We analyze both sums separately, starting with the first term. If we assume $\gamma < 1$ for now, then $\gamma^n = \sum_{i=1}^{\infty} w_i \gamma^i$ by Proposition 4. Further note that $\min(i,j) \leq (i+j) \, / \, 2$, with the inequality strict if $i \neq j$. Because $\Gamma_2$ is monotonically increasing, it follows that

$$
\begin{aligned}
\sum_{i=1}^{\infty} \sum_{j=1}^{\infty} w_i w_j \, \Gamma_2(\min(i,j)) &< \sum_{i=1}^{\infty} \sum_{j=1}^{\infty} w_i w_j \, \Gamma_2\left(\frac{i+j}{2}\right) \\
&= \sum_{i=1}^{\infty} \sum_{j=1}^{\infty} w_i w_j \left(\frac{1 - \gamma^{i+j}}{1 - \gamma^2}\right) \\
&= \frac{1 - \sum_{i=1}^{\infty} \sum_{j=1}^{\infty} w_i w_j \gamma^{i+j}}{1 - \gamma^2} \\
&= \frac{1 - \sum_{i=1}^{\infty} w_i \gamma^i \sum_{j=1}^{\infty} w_j \gamma^j}{1 - \gamma^2} \\
&= \frac{1 - \gamma^{2n}}{1 - \gamma^2} \\
&= \Gamma_2(n) \, .
\end{aligned}
$$

The inequality is strict because at least two weights in **w** are nonzero by definition of a compound return, guaranteeing at least one element in the sum has $i \neq j$. If instead $\gamma = 1$, then $n = \sum_{i=1}^{\infty} w_i i$ by Proposition 4 and also $\Gamma_2(\min(i,j)) = \min(i,j)$. Therefore, by Jensen's inequality, we have

$$
\begin{aligned}
\sum_{i=1}^{\infty} \sum_{j=1}^{\infty} w_i w_j \, \Gamma_2(\min(i,j)) &= \sum_{i=1}^{\infty} \sum_{j=1}^{\infty} w_i w_j \min(i,j) \\
&< \min\left(\sum_{i=1}^{\infty} w_i i, \ \sum_{j=1}^{\infty} w_j j\right) \\
&= \min(n, \, n) \\
&= \Gamma_2(n) \, .
\end{aligned}
$$

Again, the inequality is strict by definition of a compound return, so we conclude that

$$\sum_{i=1}^{\infty} \sum_{j=1}^{\infty} w_i w_j \, \Gamma_2(\min(i,j)) < \Gamma_2(n), \quad \text{for } 0 < \gamma \leq 1 \, .$$

We now address the second term. We show that $\Gamma_1$ is invariant under a weighted average under our assumption that Eq. (11) holds. If $\gamma < 1$, then

$$\sum_{i=1}^{\infty} w_i \, \Gamma_1(i) = \sum_{i=1}^{\infty} w_i \left(\frac{1 - \gamma^i}{1 - \gamma}\right) = \frac{1 - \sum_{i=1}^{\infty} w_i \gamma^i}{1 - \gamma} = \frac{1 - \gamma^n}{1 - \gamma} = \Gamma_1(n) \, . \qquad (17)$$

If $\gamma = 1$, then

$$\sum_{i=1}^{\infty} w_i \, \Gamma_1(i) = \sum_{i=1}^{\infty} w_i i = n = \Gamma_1(n) \, .$$

Thus, regardless of $\gamma$, the second term becomes

$$\sum_{i=1}^{\infty} \sum_{j=1}^{\infty} w_i w_j \, \Gamma_1(i) \, \Gamma_1(j) = \sum_{i=1}^{\infty} w_i \, \Gamma_1(i) \sum_{j=1}^{\infty} w_j \, \Gamma_1(j) = \Gamma_1(n)^2 \, .$$

Putting everything together, we have so far shown that

$$\text{Var}\left[G_t^{\mathbf{W}}\right] \leq (1 - \rho) \, \Gamma_2(n) \, \kappa + \rho \, \Gamma_1(n)^2 \, \kappa \, ,$$

where the right-hand side is the $n$-step return variance given by Proposition 1. As we showed above, this inequality is strict whenever the first term is active, i.e., $\rho < 1$, which completes the proof. $\qquad \square$

**Proposition 5** (Effective $\lambda$ of $n$-step return). *For any $n \geq 1$, the $\lambda$-return with $\lambda = (1 - \gamma^{n-1}) /$ $(1 - \gamma^n)$ has the same contraction rate as the $n$-step return when $\gamma < 1$, and the $\lambda$-return with $\lambda = (n - 1) / n$ has the COM as the $n$-step return when $\gamma = 1$.*

*Proof.* **Case $\gamma < 1$:** We substitute $w_n = (1 - \lambda)\lambda^{n-1}$ into the weighted average in Proposition 3 to compute the contraction rate of the $\lambda$-return:

$$\sum_{n=1}^{\infty} w_n \gamma^n = \sum_{n=1}^{\infty} (1 - \lambda)\lambda^{n-1}\gamma^n = \gamma(1 - \lambda)\sum_{n=1}^{\infty}(\gamma\lambda)^{n-1} = \frac{\gamma(1 - \lambda)}{1 - \gamma\lambda} \, .$$

We therefore seek $\lambda$ such that

$$\frac{\gamma(1 - \lambda)}{1 - \gamma\lambda} = \gamma^n$$

in order to equate the $\lambda$-return's contraction rate to that of the given $n$-step return. We multiply both sides of the equation by $1 - \gamma\lambda$ and isolate $\lambda$ to complete the case:

$$\begin{aligned}
\gamma(1 - \lambda) &= \gamma^n(1 - \gamma\lambda) \\
1 - \lambda &= \gamma^{n-1}(1 - \gamma\lambda) \\
1 - \lambda &= \gamma^{n-1} - \gamma^n\lambda \\
\gamma^n\lambda - \lambda &= \gamma^{n-1} - 1 \\
\lambda(\gamma^n - 1) &= \gamma^{n-1} - 1 \\
\lambda &= (1 - \gamma^{n-1}) / (1 - \gamma^n) \, .
\end{aligned}$$

**Case $\gamma = 1$:** We use Proposition 4 with $w_n = (1 - \lambda)\lambda^{n-1}$ to compute the effective $n$-step of the $\lambda$-return:

$$n = \sum_{n'=1}^{\infty}(1 - \lambda)\lambda^{n'-1}n' = (1 - \lambda)\sum_{n'=1}^{\infty}\lambda^{n'-1}n' = (1 - \lambda)\frac{1}{(1 - \lambda)^2} = \frac{1}{1 - \lambda} \, .$$

Rearranging the equation $n = 1 / (1 - \lambda)$ for $\lambda$ gives the final result of $\lambda = (n - 1) / n$. $\qquad\square$

## B.2  FINITE-TIME ANALYSIS

In this section, we prove Theorem 2 to establish a finite-time bound on the performance of multistep TD learning. We derive the bound in terms of the return variance, allowing us to invoke Theorem 1 and show an improved convergence rate.

At each iteration, TD learning updates the current parameters $\theta_t \in \mathbb{R}^d$ according to Eq. (12). A value-function estimate for any state $s$ is obtained by evaluating the dot product of the parameters and the state's corresponding feature vector: $V_\theta(s) \stackrel{\text{def}}{=} \theta^\top \phi(s)$. Following Bhandari et al. (2018), we assume that $\|\phi(s)\|_2^2 \leq 1, \forall\, s \in \mathcal{S}$. This can be guaranteed in practice by normalizing the features and is therefore not a strong assumption.

In the prediction setting, the agent's behavior policy is fixed such that the MDP can be cast as a Markov reward process (MRP), where $R(s, s')$ denotes the expected reward earned when transitioning from state $s$ to state $s'$. We adopt the i.i.d. state model from Bhandari et al. (2018, sec. 3) and generalize it for multistep TD updates.

**Assumption 3** (i.i.d. state model). *Assume the MRP under the policy $\pi$ is ergodic. Let $d \in \mathbb{R}^{|\mathcal{S}|}$ represent the MRP's unique stationary distribution. Each iteration of Eq. (12) is calculated by first sampling a random initial state $S_{t,0} \sim d$ and then sampling a trajectory of subsequent states $S_{t,i+1} \sim \Pr(\cdot \mid S_{t,i}), \forall\, i \in \mathbb{N}_0$.*

That is, a state $S_{t,0}$ sampled from the steady-state MRP forms the root node for the following trajectory $(S_{t,1}, S_{t,2}, \dots)$ that is generated according to the MRP transition function. Notably, this setting closely models the experience-replay setting utilized by many deep RL agents.

To facilitate our analysis, we decompose the compound TD updates into weighted averages of $n$-step TD updates, where each $n$-step update has the form

$$g_t^n(\theta) \stackrel{\text{def}}{=} \left(G_t^n - \phi(S_t)^\top \theta\right) \phi(S_t) \,.$$

This allows us to conveniently express a compound TD update as

$$g_t^{\mathbf{W}}(\theta) \stackrel{\text{def}}{=} \sum_{n=1}^\infty w_n g_t^n(\theta) \,.$$

Our proofs also make use of the *expected* $n$-step TD update:

$$\bar{g}^n(\theta) \stackrel{\text{def}}{=} \sum_{s_0 \in \mathcal{S}} \sum_{\tau \in \mathcal{S}^{n-1}} d(s_0) \Pr(\tau \mid s_0) \left(G^n(s_0, \tau, \theta) - \phi(s_0)^\top \theta\right) \phi(s_0)$$

where $(s_1, s_2, \dots) = \tau$ and $G^n(s_0, \tau, \theta) \stackrel{\text{def}}{=} R(s_0, s_1) + \cdots + \gamma^{n-1} R(s_{n-1}, s_n) + \gamma^n \phi(s_n)^\top \theta$ is the $n$-step return generated from $(s_0, \tau)$.

For brevity, let $R_i \stackrel{\text{def}}{=} R(S_{t,i}, S_{t,i+1})$ and $\phi_i \stackrel{\text{def}}{=} \phi(S_{t,i})$ be random variables sampled according to Assumption 3. We conveniently write the expected $n$-step TD update as

$$\bar{g}^n(\theta) = \mathbb{E}\left[\phi_0(R_0 + \gamma R_1 + \cdots + \gamma^{n-1} R_{n-1})\right] + \mathbb{E}\left[\phi_0(\gamma^n \phi_n - \phi_0)^\top\right] \theta \,. \tag{18}$$

The expected compound TD update easily follows as the weighted average

$$\bar{g}^{\mathbf{W}}(\theta) \stackrel{\text{def}}{=} \sum_{n=1}^\infty w_n \bar{g}^n(\theta) \,.$$

Finally, let $\theta^*$ be the fixed point of the compound TD update: i.e., $\bar{g}^{\mathbf{W}}(\theta^*) = 0$. This fixed point always exists and is unique because the projected Bellman operator is a contraction mapping (Tsitsiklis & Van Roy, 1996), and therefore so is any weighted average of the $n$-iterated operators.

Before we prove Theorem 2, we must introduce two lemmas. The first establishes a lower bound on the angle between the expected TD update and the true direction toward the fixed point.

**Lemma 2.** *Define the diagonal matrix $D \stackrel{def}{=} diag(d)$. For any $\theta \in \mathbb{R}^d$,*

$$(\theta^* - \theta)^\top \bar{g}^{\mathbf{W}}(\theta) \geq (1 - \beta) \|V_{\theta^*} - V_\theta\|_D^2 \,. \tag{19}$$

*Proof.* Let $\xi_i \stackrel{\text{def}}{=} V_{\theta^*}(S_{t,i}) - V_\theta(S_{t,i}) = (\theta^* - \theta)^\top \phi_i$ for $i \in \mathbb{N}_0$. By stationarity, each $\xi_i$ is a correlated random variable with the same marginal distribution. Because $S_{t,0}$ is drawn from the stationary distribution, we have $\mathbb{E}\left[\xi_i^2\right] = \|V_{\theta^*} - V_\theta\|_D^2$.

From Eq. (18), we show

$$\bar{g}^{\mathbf{W}}(\theta) = \bar{g}^{\mathbf{W}}(\theta) - \bar{g}^{\mathbf{W}}(\theta^*) = \sum_{n=1}^\infty w_n \mathbb{E}\left[\phi_0(\gamma^n \phi_n - \phi_0)^\top(\theta - \theta^*)\right]$$

$$= \sum_{n=1}^\infty w_n \mathbb{E}[\phi_0(\xi_0 - \gamma^n \xi_n)] \,.$$

It follows that

$$(\theta^* - \theta)^\top \bar{g}^{\mathbf{W}}(\theta) = \sum_{n=1}^\infty w_n \mathbb{E}[\xi_0(\xi_0 - \gamma^n \xi_n)]$$

$$= \mathbb{E}\left[\xi_0^2\right] - \sum_{n=1}^\infty w_n \gamma^n \mathbb{E}[\xi_0 \xi_n]$$

$$\geq \left(1 - \sum_{n=1}^\infty w_n \gamma^n\right) \mathbb{E}\left[\xi_0^2\right]$$

$$= (1 - \beta) \|V_{\theta^*} - V_\theta\|_D^2 \,.$$

The inequality uses the Cauchy-Schwartz inequality along with the fact that every $\xi_i$ has the same marginal distribution: thus, $\mathbb{E}[\xi_0 \xi_i] \leq \sqrt{\mathbb{E}[\xi_0^2]}\sqrt{\mathbb{E}[\xi_i^2]} = \mathbb{E}\left[\xi_0^2\right]$. $\quad\square$

The next lemma establishes a bound on the second moment of the squared norm of the TD update in terms of the contraction rate $\beta$ and the variance $\sigma^2$ of the compound return.

**Lemma 3.** *Define* $\Delta^* \overset{\text{def}}{=} \|R\|_\infty + (1+\gamma)\|\theta^*\|_\infty$ *and* $C \overset{\text{def}}{=} \Delta^* / (1-\gamma)$. *For any* $\theta \in \mathbb{R}^d$,

$$\mathbb{E}[\|g_t(\theta)\|_2^2] \leq 2(1-\beta)^2 C^2 + 2\sigma^2 + 4(1+\beta)\|V_{\theta^*} - V_\theta\|_D^2.$$

*Proof.* Let $\delta_{t,i}^* \overset{\text{def}}{=} R_i + \gamma\phi_{i+1}^\top\theta^* - \phi_i^\top\theta^*$ and note that $|\delta_{t,i}^*| \leq \Delta^*$ for all $i \in \mathbb{N}_0$ by the triangle inequality and the bounded-feature assumption. Denote the $n$-step and compound errors constructed from $\theta^*$ by $\delta_t^{(n)} \overset{\text{def}}{=} \sum_{i=0}^{n-1}\gamma^i\delta_{t,i}^*$ and $\delta_t^{\mathsf{w}} \overset{\text{def}}{=} \sum_{n=1}^\infty w_n\delta_t^{(n)}$, respectively. We have

$$\mathbb{E}\Big[\|g_t(\theta^*)\|_2^2\Big] = \mathbb{E}\Big[\|\delta_t^{\mathsf{w}}\phi_0\|_2^2\Big] \leq \mathbb{E}\big[(\delta_t^{\mathsf{w}})^2\big] = \mathbb{E}[\delta_t^{\mathsf{w}}]^2 + \sigma^2, \tag{20}$$

where the inequality follows from the assumption that $\|\phi_0\|_2^2 \leq 1$. The absolute value of the expectation can be bounded using the triangle inequality:

$$|\mathbb{E}[\delta_t^{\mathsf{w}}]| = \left|\mathbb{E}\left[\sum_{n=1}^\infty w_n\delta_t^{(n)}\right]\right| \leq \sum_{n=1}^\infty w_n\,\Gamma_1(n)\,\Delta^* = \frac{1-\beta}{1-\gamma}\Delta^* = (1-\beta)C. \tag{21}$$

The identity $\sum_{n=1}^\infty w_n\Gamma_1(n) = (1-\beta) / (1-\gamma)$ comes from Eq. (17). Eqs. (20) and (21) imply

$$\mathbb{E}\Big[\|g_t(\theta^*)\|_2^2\Big] \leq (1-\beta)^2 C^2 + \sigma^2. \tag{22}$$

Recall that $\mathbb{E}\big[\xi_i^2\big] = \|V_{\theta^*} - V_\theta\|_D^2$ for all $i \in \mathbb{N}_0$. Next, we show

$$\begin{aligned}
\mathbb{E}\Big[\|g_t(\theta) - g_t(\theta^*)\|_2^2\Big] &= \mathbb{E}\left[\left\|\sum_{n=1}^\infty w_n\phi_0(\gamma^n\phi_n - \phi_0)^\top(\theta - \theta^*)\right\|_2^2\right] \\
&= \mathbb{E}\left[\left\|\sum_{n=1}^\infty w_n\phi_0(\xi_0 - \gamma^n\xi_n)\right\|_2^2\right] \\
&\leq \sum_{n=1}^\infty w_n\mathbb{E}\Big[\|\phi_0(\xi_0 - \gamma^n\xi_n)\|_2^2\Big] \\
&\leq \sum_{n=1}^\infty w_n\mathbb{E}\big[(\xi_0 - \gamma^n\xi_n)^2\big] \\
&\leq 2\sum_{n=1}^\infty w_n\left(\mathbb{E}\big[\xi_0^2\big] + \gamma^{2n}\mathbb{E}\big[\xi_n^2\big]\right) \\
&= 2\sum_{n=1}^\infty w_n(1 + \gamma^{2n})\|V_{\theta^*} - V_\theta\|_D^2 \\
&\leq 2\sum_{n=1}^\infty w_n(1 + \gamma^n)\|V_{\theta^*} - V_\theta\|_D^2 \\
&= 2(1+\beta)\|V_{\theta^*} - V_\theta\|_D^2. \tag{23}
\end{aligned}$$

The four inequalities respectively follow from Jensen's inequality, the bounded-feature assumption $\|\phi\|_2^2 \leq 1$, the triangle inequality, and the fact that $\gamma^{2n} \leq \gamma^n$. The final equality comes from the definition of the contraction rate (Proposition 3). Combining Eqs. (22) and (23) gives the final result:

$$\begin{aligned}
\mathbb{E}\Big[\|g_t(\theta)\|_2^2\Big] &\leq \mathbb{E}\Big[(\|g_t(\theta^*)\|_2 + \|g_t(\theta) - g_t(\theta^*)\|_2)^2\Big] \\
&\leq 2\mathbb{E}\Big[\|g_t(\theta^*)\|_2^2\Big] + 2\mathbb{E}\Big[\|g_t(\theta) - g_t(\theta^*)\|_2^2\Big] \\
&\leq 2(1-\beta)^2 C^2 + 2\sigma^2 + 4(1+\beta)\|V_{\theta^*} - V_\theta\|_D^2,
\end{aligned}$$

where the second inequality uses the algebraic identity $(x + y)^2 \leq 2x^2 + 2y^2$. $\qquad\square$

We are now ready to derive the finite-time bound. We restate Theorem 2 and then provide the proof.

**Theorem 2** (Finite-Time Analysis of Compound TD Learning). *Suppose TD learning with linear function approximation is applied under an i.i.d. state model (see Assumption 3 in Appendix B.2) using the compound return estimator $G_t^{\mathbf{w}}$ as its target. Let $\beta \in [0, 1)$ be the contraction rate of the estimator (see Proposition 3), and let $\sigma^2 \geq 0$ be the variance of the estimator under Assumptions 1 and 2. Assume that the features are normalized such that $\|\phi(s)\|_2^2 \leq 1$, $\forall\, s \in \mathcal{S}$. Define $C \stackrel{def}{=} (\|R\|_\infty + (1 + \gamma)\|\theta^*\|_\infty) / (1 - \gamma)$, where $\theta^*$ is the minimizer of the projected Bellman error for $G_t^{\mathbf{w}}$. For any $T \geq (4 / (1 - \beta))^2$ and a constant step size $\alpha = 1 / \sqrt{T}$,*

$$\mathbb{E}\left[\left\|V_{\theta^*} - V_{\bar{\theta}_T}\right\|_D^2\right] \leq \frac{\|\theta^* - \theta_0\|_2^2 + 2(1 - \beta)^2 C^2 + 2\sigma^2}{(1 - \beta)\sqrt{T}}, \quad \text{where } \bar{\theta}_T \stackrel{def}{=} \frac{1}{T} \sum_{t=0}^{T-1} \theta_t.$$

*Proof.* TD learning updates the parameters according to Eq. (12). Therefore,

$$
\begin{aligned}
\|\theta^* - \theta_{t+1}\|_2^2 &= \|\theta^* - \theta_t - \alpha\, g_t(\theta_t)\|_2^2 \\
&= \|\theta^* - \theta_t\|_2^2 - 2\alpha\, g_t(\theta_t)^\top(\theta^* - \theta_t) + \alpha^2 \|g_t(\theta)\|_2^2.
\end{aligned}
$$

Taking the expectation and then applying Lemmas 2 and 3 gives

$$
\begin{aligned}
\mathbb{E}\left[\|\theta^* - \theta_{t+1}\|_2^2\right] &= \mathbb{E}\left[\|\theta^* - \theta_t\|_2^2\right] - 2\alpha\mathbb{E}\left[g_t(\theta_t)^\top(\theta^* - \theta_t)\right] + \alpha^2\mathbb{E}\left[\|g_t(\theta)\|_2^2\right] \\
&= \mathbb{E}\left[\|\theta^* - \theta_t\|_2^2\right] - 2\alpha\mathbb{E}\left[\mathbb{E}\left[g_t(\theta_t)^\top(\theta^* - \theta_t)\right] \mid \theta_t\right] + \alpha^2\mathbb{E}\left[\mathbb{E}\left[\|g_t(\theta)\|_2^2\right] \mid \theta_t\right] \\
&\leq \mathbb{E}\left[\|\theta^* - \theta_t\|_2^2\right] - \left(2\alpha(1 - \beta) - 4\alpha^2(1 + \beta)\right)\|V_{\theta^*} - V_\theta\|_D^2 + 2\alpha^2\left((1 - \beta)^2 C^2 + \sigma^2\right) \\
&\leq \mathbb{E}\left[\|\theta^* - \theta_t\|_2^2\right] - \alpha(1 - \beta)\|V_{\theta^*} - V_\theta\|_D^2 + 2\alpha^2\left((1 - \beta)^2 C^2 + \sigma^2\right).
\end{aligned}
$$

The first inequality is due to Lemmas 2 and 3, which are applicable due to the i.i.d. setting (because the trajectory influencing $g_t$ is independent of $\theta_t$). The second inequality follows from the assumption that $\alpha \leq (1 - \beta) / 4$. Rearranging the above inequality gives us

$$\mathbb{E}\left[\|V_{\theta^*} - V_{\theta_t}\|_D^2\right] \leq \frac{\|\theta^* - \theta_t\|_2^2 - \|\theta^* - \theta_{t+1}\|_2^2 + 2\alpha^2\left((1 - \beta)^2 C^2 + \sigma^2\right)}{\alpha(1 - \beta)}.$$

Summing over $T$ iterations and then invoking the assumption that $\alpha = 1 / \sqrt{T}$:

$$
\begin{aligned}
\sum_{t=0}^{T-1} \mathbb{E}\left[\|V_{\theta^*} - V_{\theta_t}\|_D^2\right] &\leq \frac{\|\theta^* - \theta_0\|_2^2 - \|\theta^* - \theta_T\|_2^2 + 2\alpha^2\left((1 - \beta)^2 C^2 + \sigma^2\right)T}{\alpha(1 - \beta)} \\
&\leq \frac{\|\theta^* - \theta_0\|_2^2 + 2\alpha^2\left((1 - \beta)^2 C^2 + \sigma^2\right)T}{\alpha(1 - \beta)} \\
&= \frac{\|\theta^* - \theta_0\|_2^2\sqrt{T} + 2\left((1 - \beta)^2 C^2 + \sigma^2\right)\sqrt{T}}{1 - \beta}.
\end{aligned}
$$

We therefore conclude that

$$\mathbb{E}\left[\left\|V_{\theta^*} - V_{\bar{\theta}_T}\right\|_D^2\right] \leq \frac{1}{T}\sum_{t=0}^{T-1}\mathbb{E}\left[\|V_{\theta^*} - V_{\theta_t}\|_D^2\right] \leq \frac{\|\theta^* - \theta_0\|_2^2 + 2(1 - \beta)^2 C^2 + 2\sigma^2}{(1 - \beta)\sqrt{T}},$$

which completes the bound. $\qquad\square$

## C  $\lambda$-RETURN VARIANCE

We calculate the variance of the $\lambda$-return under Assumptions 1 and 2. In the main text, we showed that the $\lambda$-return assigns a cumulative weight of $W_i = \lambda^i$ to the TD error at time $t + i$, which is also known from the TD($\lambda$) algorithm. We can therefore apply Proposition 2 to obtain the following variance expression:

$$
\begin{aligned}
\mathrm{Var}[G_t^\lambda] &= (1 - \rho) \sum_{i=0}^{\infty} (\gamma\lambda)^{2i} \kappa + \rho \sum_{i=0}^{\infty} \sum_{j=0}^{\infty} (\gamma\lambda)^{i+j} \kappa \\
&= (1 - \rho) \sum_{i=0}^{\infty} (\gamma\lambda)^{2i} \kappa + \rho \sum_{i=0}^{\infty} (\gamma\lambda)^{i} \sum_{j=0}^{\infty} (\gamma\lambda)^{j} \kappa \\
&= \frac{(1 - \rho)\kappa}{1 - (\gamma\lambda)^2} + \frac{\rho\kappa}{(1 - \gamma\lambda)^2} \; .
\end{aligned}
\tag{24}
$$

# D PiLaR: Piecewise $\lambda$-Return

We present a basic search algorithm for finding the best PiLaR for a given $n$-step return when rewards are discounted (see Algorithm 1). The algorithm accepts the desired effective $n$-step (which does not need to be an integer necessarily) as its only argument and returns the values $(n_1, n_2, w)$ such that the compound return $(1 - w)G_t^{n_1} + wG_t^{n_2}$ minimizes the maximum absolute difference between its cumulative weights and those of the $\lambda$-return with the same effective $n$-step. The algorithm proceeds as follows. For each $n_1 \in \{1, \ldots, \lfloor n \rfloor\}$, scan through $n_2 \in \{n_1 + 1, n_1 + 2, \ldots, \}$ until the error stops decreasing. Every time a better $(n_1, n_2)$-pair is found, record the values, and return the last recorded values upon termination. The resulting PiLaR has the same contraction rate as the targeted $n$-step return; thus, their error-reduction properties are the same, but the compound return's variance is lower by Theorem 1.

Table 2: $n$-step returns and PiLaRs with equal contraction rates when $\gamma = 0.99$.

| effective $n$-step | $n_1$ | $n_2$ | $w$ |
|---|---|---|---|
| 2 | 1 | 4 | 0.337 |
| 3 | 1 | 6 | 0.406 |
| 4 | 2 | 7 | 0.406 |
| 5 | 2 | 9 | 0.437 |
| 10 | 4 | 16 | 0.515 |
| 20 | 6 | 35 | 0.519 |
| 25 | 8 | 43 | 0.530 |
| 50 | 13 | 79 | 0.640 |
| 100 | 22 | 147 | 0.760 |

We populate Table 2 with corresponding PiLaR values for several common $n$-step returns when $\gamma = 0.99$. A discount factor of $\gamma = 0.99$ is extremely common in deep RL, and so it is hoped that this table serves as a convenient reference that helps practitioners avoid redundant searches.

To modify the search algorithm for undiscounted rewards, we just need to change the weight $w$ such that it equates the COMs—rather than the contraction rates—of the two returns. That is, our choice of $w$ must instead satisfy $(1 - w)n_1 + wn_2 = n$, and so it follows that $w = (n - n_1) / (n_2 - n_1)$. We make this change on line 9 and then just substitute $\gamma = 1$ elsewhere in the pseudocode to complete the search algorithm, which we include in Algorithm 2 for a side-by-side comparison with the discounted setting.

---

**Algorithm 1** PiLaR($n$)    ($0 < \gamma < 1$)

1: **require** $n \geq 1$
2: $\lambda = (1 - \gamma^{n-1}) / (1 - \gamma^n)$
3: best_error $\leftarrow \infty$
4: **for** $n_1 = 1, \ldots, \lfloor n \rfloor$ **do**
5:   $n_2 \leftarrow \lfloor n \rfloor$
6:   error $\leftarrow \infty$
7:   **repeat**
8:     $n_2 \leftarrow n_2 + 1$
9:     $w \leftarrow (\gamma^n - \gamma^{n_1}) / (\gamma^{n_2} - \gamma^{n_1})$
10:     prev_error $\leftarrow$ error
11:     error $\leftarrow$ ERROR($\lambda, n_1, n_2, w$)
12:     **if** error $<$ best_error **then**
13:       values $\leftarrow (n_1, n_2, w)$
14:       best_error $\leftarrow$ error
15:     **end if**
16:   **until** error $\geq$ prev_error
17: **end for**
18: **return** values

19: **function** ERROR($\lambda, n_1, n_2, w$)
20:   Let $W_i = \begin{cases} \gamma^i & \text{if } i < n_1 \\ w\gamma^i & \text{else if } i < n_2 \\ 0 & \text{else} \end{cases}$
21:   **return** $\max_{i \geq 0} \left| W_i - (\gamma\lambda)^i \right|$
22: **end function**

---

**Algorithm 2** PiLaR($n$)    ($\gamma = 1$)

1: **require** $n \geq 1$
2: $\lambda = (n - 1) / n$
3: best_error $\leftarrow \infty$
4: **for** $n_1 = 1, \ldots, \lfloor n \rfloor$ **do**
5:   $n_2 \leftarrow \lfloor n \rfloor$
6:   error $\leftarrow \infty$
7:   **repeat**
8:     $n_2 \leftarrow n_2 + 1$
9:     $w \leftarrow (n - n_1) / (n_2 - n_1)$
10:     prev_error $\leftarrow$ error
11:     error $\leftarrow$ ERROR($\lambda, n_1, n_2, w$)
12:     **if** error $<$ best_error **then**
13:       values $\leftarrow (n_1, n_2, w)$
14:       best_error $\leftarrow$ error
15:     **end if**
16:   **until** error $\geq$ prev_error
17: **end for**
18: **return** values

19: **function** ERROR($\lambda, n_1, n_2, w$)
20:   Let $W_i = \begin{cases} 1 & \text{if } i < n_1 \\ w & \text{else if } i < n_2 \\ 0 & \text{else} \end{cases}$
21:   **return** $\max_{i \geq 0} \left| W_i - \lambda^i \right|$
22: **end function**

# E  EXPERIMENT SETUP AND ADDITIONAL RESULTS

Our experiment procedure closely matches that of Young & Tian (2019) for DQN. The only differences in our methodology are the following: dueling network architecture, Adam optimizer, minibatch updates every 4 steps (instead of 1), and multistep returns.

MinAtar represents states as $10 \times 10 \times 7$ binary images. The agent processes these with a convolutional network; the first layer is a 16-filter $3 \times 3$ convolutional layer, the output of which is flattened and then followed by a dense layer with 128 units. Both layers use ReLU activations.

We further adopt the dueling network architecture from Wang et al. (2016), which we found to help value estimation significantly. Rather than directly mapping the 128 extracted features to a vector of action-values with a linear layer, as is normally done by DQN, the dueling network splits the output into two linear heads: a scalar value estimate $V(s; \theta_V)$ and a vector of advantage estimates $A(s, \cdot; \theta_A)$, where $\theta_V, \theta_A \subset \theta$. The action-value $Q(s, a)$ is then obtained by

$$Q(s, a; \theta) = V(s; \theta_V) + A(s, a; \theta_A) - \max_{a' \in \mathcal{A}} A(s, a'; \theta_A). \tag{25}$$

With automatic differentiation software, this change is transparent to the agent. Note that here we are subtracting $\max_{a' \in \mathcal{A}} A(s, a'; \theta_A)$ for the Bellman identifiability term, which is the less common but more principled formulation for predicting $q_*$ (Wang et al., 2016, eq. 8). We found this to work better than the typical subtraction by $(1 / |\mathcal{A}|) \sum_{a' \in \mathcal{A}} A(s, a'; \theta_A)$.

The agents trained for 5 million time steps each. They executed a random policy for the first 5k time steps to prepopulate the replay buffer (capacity: 100k transitions), and then switched to an $\epsilon$-greedy policy for the remainder of training, with $\epsilon$ annealed linearly from 1 to 0.1 over the next 100k steps. Every 4 steps, the main network was updated with a minibatch of 32 return estimates to minimize the loss from Section 5. The target network's parameters were copied from the main network every 1k time steps.

To obtain the $n$-step returns, the replay buffer is modified to return a minibatch of sequences of $n + 1$ experiences for each return estimate (instead of the usual 2 experiences for DQN). The return is computed by summing the first $n$ rewards and then adding the value-function bootstrap from the final experience, with discounting if $\gamma < 1$. If the episode terminates at any point within this trajectory, then the return is truncated and no bootstrapping is necessary, since the value of a terminal state is defined to be zero. For PiLaRs, the idea is the same, but the trajectories must have length $n_2 + 1$ to accommodate the lengths of both $n$-step returns. The two returns are computed as above, and then combined by averaging them: $(1 - w)G^{n_1} + wG^{n_2}$.

We show the learning curves for the best step size found for each return estimator in Figure 6, which correspond to the horizontal dashed lines in the $\alpha$-sweep plots in Figure 3 in the main paper. We observe that PiLaRs always perform better than $n$-step returns in this environment. The $n$-step return becomes noticeably unstable when $n = 10$ due to the higher variance, which widens the relative improvement of PiLaR as it tolerates the variance better.

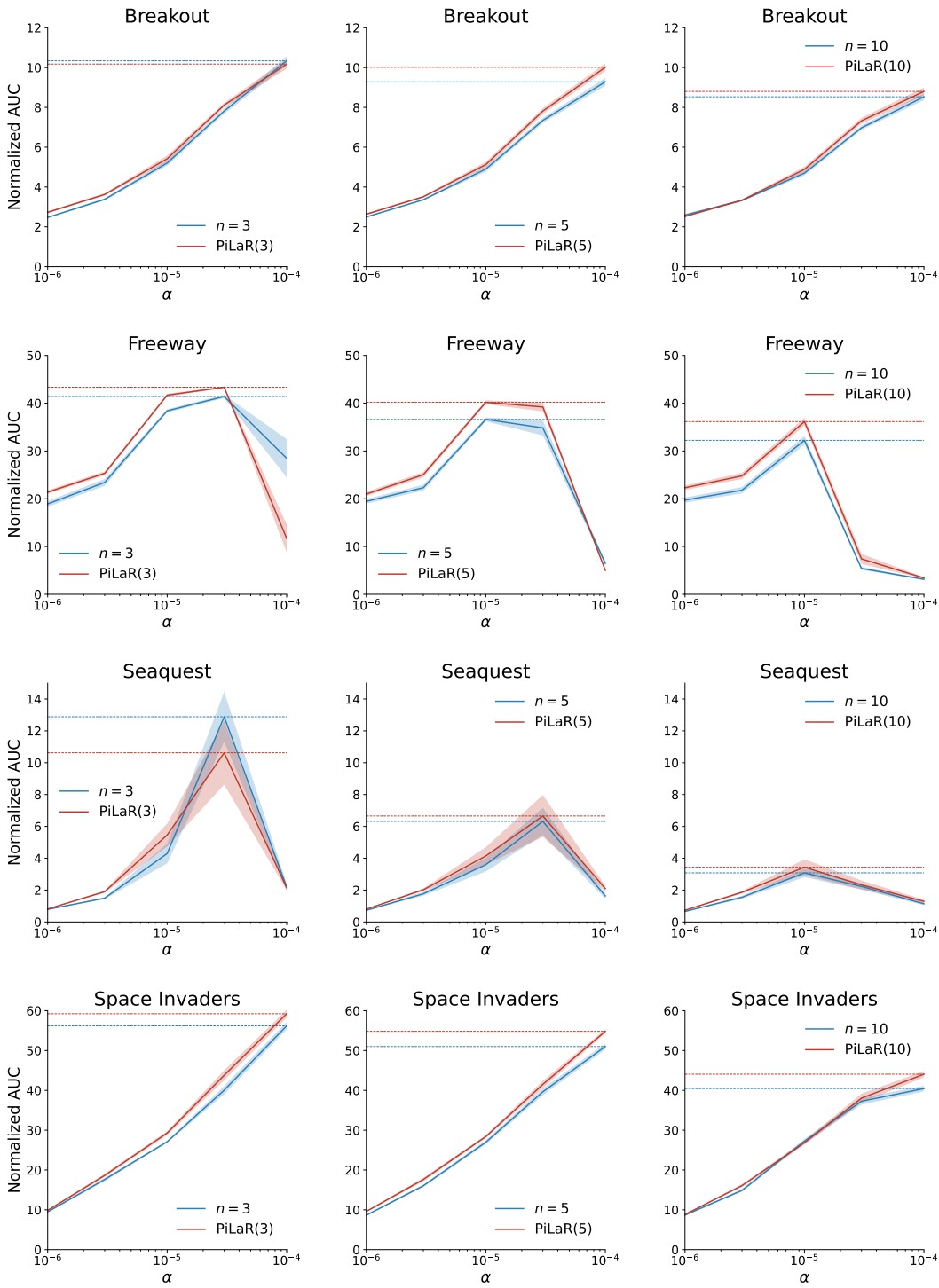

Figure 5: Step-size sensitivity curves for $n$-step returns and PiLaRs in four MinAtar games.

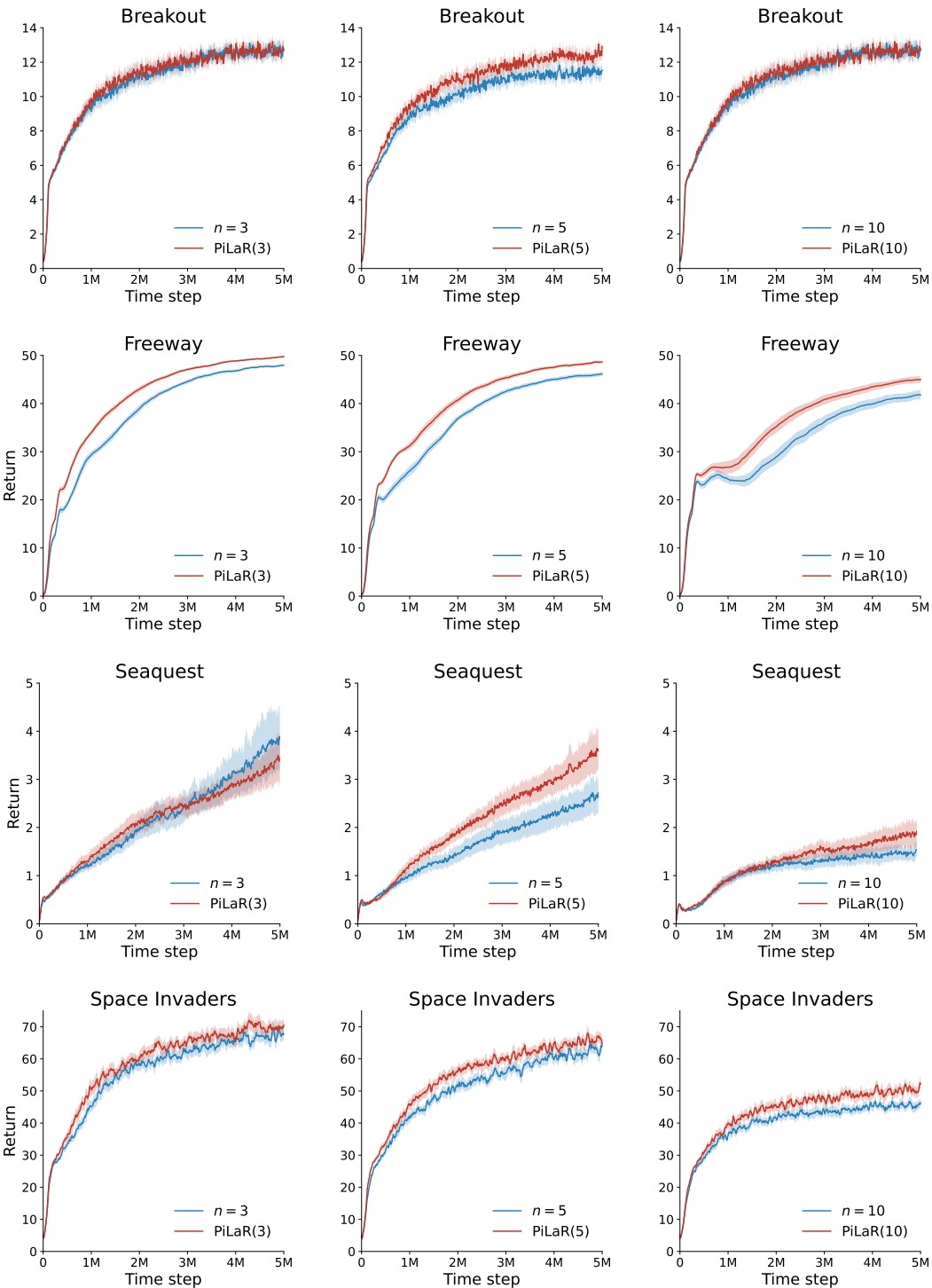

Figure 6: The best learning curves for each return estimator tested in four MinAtar games.

