# OpenReview forum: "Compound Returns Reduce Variance in Reinforcement Learning"
_ICLR.cc/2024/Conference — Submitted to ICLR 2024_

### Official Review · Reviewer_fKNs · 2023-10-17

**Soundness:** 2 fair
**Presentation:** 3 good
**Contribution:** 2 fair
**Rating:** 5
**Confidence:** 3

**Summary:**

This paper analyzes the variance reduction property of compound returns. While compound returns, such as the lambda-return, are often viewed as helping with variance reduction via averaging, the authors claim that this variance properties is formally investigate for the first time. Under certain assumptions on the variance/covariance model, the authors prove for the first time that any compound return with the same contraction rate as a given n-step return has strictly lower variance. The studies shed light on the theoretical understanding of using compound returns in learning value functions. Subsequently, the authors propose a computationally friendly piecewise lambda-return and verify the efficacy of their approach on one Atari Freeway environment.

**Strengths:**

The paper considers an interesting question. While it may be commonly believed that averaging helps with variance and hence learning in RL, the authors formally study the problem and show that compound returns admit a better bias-variance trade-off. The writing is overall very clear and organized. The proposed piecewise lambda-return is theoretically sound and seems to also perform in the limited experimental evaluations.

**Weaknesses:**

While a formal study on the variance reduction property is valuable, the theoretical contributions of this paper seem limited. The assumptions help abstract a lot of the difficulties and with the uniform variance/correlation assumptions, the derivation in this paper seems to be straightforward/follow standard arguments. As such, the technical depth is limited. Consequently, for such paper with limited theoretical innovations, one might expect a more comprehensive experimental evaluations, ablation studies and comparisons. The current manuscript unfortunately only evaluates on the Atari Freeway environment.

**Questions:**

See weaknesses.

---

> ### Author Response · Authors · 2023-11-16
> **Official Response to Reviewer fkNs**
>
> Thank you for the review. We respond to each of your concerns below:
>
> > While a formal study on the variance reduction property is valuable, the theoretical contributions of this paper seem limited. The assumptions help abstract a lot of the difficulties and with the uniform variance/correlation assumptions, the derivation in this paper seems to be straightforward/follow standard arguments. As such, the technical depth is limited.
>
> Our work is the first to make progress on this question in the last 12 years. Please see “**Strength of Theoretical Contributions**” in our [general response](https://openreview.net/forum?id=f37TVPH62h&noteId=TJYI9LMG6P) for more details about the novelty of our theory.
>
> > One might expect a more comprehensive experimental evaluations, ablation studies and comparisons. The current manuscript unfortunately only evaluates on the Atari Freeway environment.
>
> Thank you for this suggestion. We have added results for three more MinAtar games to the paper; please see “**Additional Experiments**” in our [general response](https://openreview.net/forum?id=f37TVPH62h&noteId=GK2F7B8JpD).

---

### Official Review · Reviewer_Ucxg · 2023-10-26

**Soundness:** 3 good
**Presentation:** 2 fair
**Contribution:** 2 fair
**Rating:** 3
**Confidence:** 3

**Summary:**

This paper analyzes the widely used $\lambda$-compounded returns and show that they have lower variance than $n$-steps return if the temporal difference errors have equal correlation strictly less than one.

In addition they propose PILAR which is a practical deep RL approximation of the TD($\lambda$) compatible with experience replay.

**Strengths:**

The paper discovers few new characteristics of a very common method in RL that is TD($\lambda$).

**Weaknesses:**

1) I think that most of the results are slightly incremental and not clearly novel.

2) Assuming that all temporal differences error have the same correlation is a strong assumption in my opinion.

3) In general in RL it is not clear if minimizing the variance of the return estimators is helpful to improve the sample complexity of an algorithm. Check for example this paper investigating the role of the minimum variance baseline in policy gradient as in [1].

4) The experiments in Deep RL are limited to only one environment. I think that a larger empirical evaluation is necessary.

[1] Wesley Chung, Valentin Thomas, Marlos C Machado, and Nicolas Le Roux.
Beyond variance reduction: Understanding the true impact of baselines on policy optimization

**Questions:**

Is it possible to use the results in this paper to show more informative results regarding the performance of TD($\lambda$). For example that having a lower variance in the returns improves the sample complexity needed for either policy evaluation or for learning an $\epsilon$-optimal policy ?

---

> ### Author Response · Authors · 2023-11-16
> **Official Response to Reviewer Ucxg**
>
> Thank you for the review. We would like to note that our theoretical analysis extends far beyond just TD($\lambda$) and $\lambda$-returns. Our analysis encompasses all arbitrary weighted averages of $n$-step returns, which is a vast family of return estimators used by all value-based RL methods. TD($\lambda$) and $\lambda$-returns are just one interesting example because of their widespread use in RL.
>
> We also want to emphasize that, until now, no one has shown that TD($\lambda$) reduces variance compared to $n$-step TD. It was previously believed [1, ch. 12.1] that the main benefit of $\lambda$-returns was their convenient implementation using eligibility traces, and were otherwise equivalent to $n$-step returns.
>
> We respond to each of your concerns below:
>
> > I think that most of the results are slightly incremental and not clearly novel. Assuming that all temporal difference errors have the same correlation is a strong assumption in my opinion.
>
> Our work is the first to make progress on this question in the last 12 years. Please see “**Strength of Theoretical Contributions**” in our [general response](https://openreview.net/forum?id=f37TVPH62h&noteId=TJYI9LMG6P) for more details about the novelty of our theory.
>
> > In general in RL it is not clear if minimizing the variance of the return estimators is helpful to improve the sample complexity of an algorithm. Check for example this paper investigating the role of the minimum variance baseline in policy gradient as in [1].
>
> This is an excellent point. It is true that reducing the return variance is not guaranteed to improve the sample efficiency of *policy gradient* methods. The paper that you cited shows that the variance of the *baseline* can sometimes be beneficial because of its interaction with the exploration policy (what the authors call committal and non-committal baselines). Our value-based setting is fundamentally different because it does not rely on a critic to determine which actions should be reinforced. Instead, the policy is directly determined as a function of the value function (e.g., $\epsilon$-greedy) and so the main bottleneck for learning is how quickly the returns can be estimated accurately.
>
> Furthermore, it is widely accepted that variance reduction is beneficial for *learning value functions*. This is why TD-based returns (e.g., n-step returns [2] or lambda-returns [3]) are almost always used in deep RL instead of Monte Carlo returns, as the latter have extremely high variance. To isolate the effects of return estimation itself, we chose to focus on DQN, a well-studied deep RL method that only learns a value function. The main benefit of our method, PiLaR, is to apply similar variance reduction in off-policy experience-replay settings where computing the full $\lambda$-return is infeasible.
>
> > The experiments in Deep RL are limited to only one environment. I think that a larger empirical evaluation is necessary.
>
> Thank you for this suggestion. We have added results for more MinAtar games to the paper; please see “**Additional Experiments**” in our [general response](https://openreview.net/forum?id=f37TVPH62h&noteId=GK2F7B8JpD).
>
> > Is it possible to use the results in this paper to show more informative results regarding the performance of TD($\lambda$). For example that having a lower variance in the returns improves the sample complexity needed for either policy evaluation or for learning an $\epsilon$-optimal policy?
>
> This is a great question. Please see “**Why Variance Reduction Leads to Faster Learning**” in our [general response](https://openreview.net/forum?id=f37TVPH62h&noteId=GK2F7B8JpD) for a detailed discussion of how the contraction rate corresponds to expected policy improvement in Q-Learning methods. We have also added this discussion to our paper’s appendix.
>
>
> **References**
>
> [1] Reinforcement Learning: An Introduction. Sutton and Barto, 2018.
>
> [2] Rainbow: Combining Improvements in Deep Reinforcement Learning. Matteo Hessel et al., AAAI 2018.
>
> [3] High-Dimensional Continuous Control Using Generalized Advantage Estimation. John Schulman et al., ICLR 2016.

---

> > ### Comment · Reviewer_Ucxg · 2023-11-17
> >
> > Thanks for your answer and for adding new experiments.
> >
> > Unfortunately, I feel that the discussion regarding “Why Variance Reduction Leads to Faster Learning” should be formalised better to push the paper at the level of acceptance.
> >
> > For example, could you combine the novel variance bound with the techniques in [1] and compare the result you would obtained with the bounds given in [1, Theorem 4] ?
> >
> > Best,
> >
> > Reviewer Ucxg
> >
> >
> > [1]   A Finite Time Analysis of Temporal Difference Learning With Linear Function Approximation    https://arxiv.org/pdf/1806.02450.pdf

---

> > > ### Author Response · Authors · 2023-11-22
> > > **Finite-Time Analysis**
> > >
> > > Thank you for your reply. We agree that the paper is much stronger with a finite-time analysis clearly showing that variance reduction leads to faster learning.
> > >
> > > - **We added a theorem (Section 3.2) deriving a sample complexity bound for any compound return** by extending the 1-step TD analysis from the paper you cited. (Note that we extended theorem 2 and not theorem 4 because our paper does not use eligibility traces.)
> > > - **The full proof has been added as Appendix B.2.**
> > > - All changes related to finite-time analysis are highlighted in blue text.
> > >
> > > Our bound contains several terms that depend on the return's contraction rate $\beta$, and one term that depends on the return's variance $\sigma^2$. **Thus, given an $n$-step return and a compound return with the same contraction rate, the compound return has lower variance (Theorem 1 in our paper) and therefore converges faster to its respective TD fixed point.**
> > >
> > > We hope that this addresses your remaining concerns about variance reduction. Please let us know if you have any questions.

---

> > > > ### Comment · Reviewer_Ucxg · 2023-11-22
> > > >
> > > > dear authors,
> > > >
> > > > thanks a lot for implementing this change! i think they go in the right direction.
> > > >
> > > > I am not be able to check the proof before the end of the reviewer authors discuss so I am keeping my original score for th time being. However I will check the proof in the AC reviewer discuss phase.
> > > >
> > > > best,
> > > >
> > > > reviewer

---

> > > > > ### Author Response · Authors · 2023-11-23
> > > > >
> > > > > Thank you! We appreciate the extra time and consideration for the proof.

---

> > > > > > ### Comment · Reviewer_Ucxg · 2023-12-03
> > > > > >
> > > > > > Dear Authors,
> > > > > >
> > > > > > I have inspected the proof. While it is well written and correct, I think that Theorem 2 leaves open the question of which is the best value of n to minimize the policy evaluation error at the fastest rate. For this reason, I feel that the paper is incomplete and could be further improved with some more effort.
> > > > > >
> > > > > > Sorry for the disappointing news. Still hope that my suggestions gave you ideas for your next steps !
> > > > > >
> > > > > > Best,
> > > > > > Reviewer

---

### Official Review · Reviewer_U4yb · 2023-11-01

**Soundness:** 2 fair
**Presentation:** 4 excellent
**Contribution:** 3 good
**Rating:** 5
**Confidence:** 3

**Summary:**

This paper studies the variance of compound returns in reinforcement learning both theoretically and empirically. Under the uniform covariance assumption between TD error at different steps, it proves that any compound return has lower variance than corresponding $n$-step return with the same contraction rate as long as the TD errors are not perfectly correlated. The contraction rate measures that convergence speed of a $n$-step TD estimator for value function estimation of a policy. Therefore, it concludes that compound return in general has lower variance under the same convergence rate. They also conduct experiments to verify this effect in value estimation tasks. Empirically, the paper proposes an approximation of $\lambda$-return using only the mixture of two multi-step returns named Piecewise $\lambda$-Return (PiLaR). Experiments with DQN on a tabular example shows the effectiveness of PiLaR on top of the standard $n$-step TD learning.

**Strengths:**

The variance of compound returns is a fundamental problem in reinforcement learning. This paper provides new insights on this problem by verifying the compound returns have lower variance than $n$-step returns under uniform covariance assumption. The paper clearly lists convincing theoretical and empirical evidence to support this claim.

**Weaknesses:**

1. It is unclear whether the uniform covariance assumption is reasonable in real-world problems, since the hardness to approximate the covariance between different steps should not be an evidence to support the validity of this assumption. Intuitively, the variance of TD errors at further steps should be larger since the entropy of state should increase along the diffusion over the MDP. Therefore, it is appreciated to verify this assumption empirically on synthetic examples.

2. The contraction rate measures the contraction level of the policy evaluation process. It is not clear the effect of this rate in the policy optimization process, nor is it discussed in the paper. Therefore, it is still not clear whether the faster learning with DQN is a consequence of smaller variance of PiLaR or smaller contraction rate in the policy optimization process as $n_2$ is generally larger than $n$.

3. The theoretical results of the paper are mostly conceptual in the sense that it proves some variance reduction results but do not discuss how this lower variance accelerate the learning of optimal policies. The "equally fast" claim for two estimators with the same contraction rate is also conceptual without solid evidence. Does it correspond to smaller sample complexity in theory? The insight of this paper is also limited in both practice and theory, since the baseline is the $n$-step TD learning and DQN, which is away from current SOTA algorithms used in RL. Is is possible to compare the PiLaR (or more refined compound error with even smaller variances) with some SOTA RL algorithms such PPO or CQL?

**Questions:**

See above.

Eqn. (8): the second $S_t$ --> $s$

Eqn. (12): missing $\kappa$ in the RHS

---

> ### Author Response · Authors · 2023-11-16
> **Official Response to Reviewer U4yb**
>
> Thank you for the review and for pointing out the two typos in our equations. We have fixed them in the paper. We also respond to each of your concerns below:
>
> > It is unclear whether the uniform covariance assumption is reasonable in real-world problems, since the hardness to approximate the covariance between different steps should not be an evidence to support the validity of this assumption. Intuitively, the variance of TD errors at further steps should be larger since the entropy of state should increase along the diffusion over the MDP. Therefore, it is appreciated to verify this assumption empirically on synthetic examples.
>
> Our work is the first to make progress on this question in the last 12 years. Please see “**Strength of Theoretical Contributions**” in our [general response](https://openreview.net/forum?id=f37TVPH62h&noteId=TJYI9LMG6P) for a detailed discussion.
>
> We think it is a great idea to measure how well our assumptions hold in practice. We added two experiments to the paper (Appendix B) that plots the actual variance of the n-step returns compared to the lower and upper bounds predicted by our model.
>
> > The contraction rate measures the contraction level of the policy evaluation process. It is not clear the effect of this rate in the policy optimization process, nor is it discussed in the paper. Therefore, it is still not clear whether the faster learning with DQN is a consequence of smaller variance of PiLaR or smaller contraction rate in the policy optimization process as $n2$ is generally larger than $n$.
>
> Thank you for bringing up this point. Please see “**Why Variance Reduction Leads to Faster Learning**” in our [general response](https://openreview.net/forum?id=f37TVPH62h&noteId=GK2F7B8JpD) for a detailed discussion of how the contraction rate corresponds to expected policy improvement in Q-Learning methods.
>
> Furthermore, it can be seen from the MinAtar experiments (Figures 5 and 6) that a larger value of $n$ actually hurts performance due to the increased variance, so it is not true that a larger $n_2$ value for PiLaR is responsible for the faster learning. Since each pair of returns are chosen to have the same contraction rate, the only remaining explanation is the reduced variance, which is supported both by our theory and random walk experiments in Figure 1.
>
> > The theoretical results of the paper are mostly conceptual in the sense that it proves some variance reduction results but do not discuss how this lower variance accelerates the learning of optimal policies. The "equally fast" claim for two estimators with the same contraction rate is also conceptual without solid evidence. Does it correspond to smaller sample complexity in theory?
>
> This is a great question. We would first like to point out that our random walk experiment in Figure 1 does provide solid evidence that two returns with the same contraction rate learn equally fast in expectation (small step size) but the one with lower variance learns faster with limited samples (large step size). Again, please see “**Why Variance Reduction Leads to Faster Learning**” in our [general response](https://openreview.net/forum?id=f37TVPH62h&noteId=GK2F7B8JpD) for answers to your questions regarding contraction rate and sample complexity.
>
> > The insight of this paper is also limited in both practice and theory, since the baseline is the n-step TD learning and DQN, which is away from current SOTA algorithms used in RL. Is it possible to compare the PiLaR (or more refined compound error with even smaller variances) with some SOTA RL algorithms such PPO or CQL?
>
> The principal contribution of our paper is the theory. The purpose of our experiments is to test how well our theory is reflected in practice when the assumptions required for the math do not always hold, as well as to demonstrate the feasibility of using variance-reducing compound returns in deep RL. We further discuss this in “**Additional Experiments**” in our [general response](https://openreview.net/forum?id=f37TVPH62h&noteId=GK2F7B8JpD), where we provide results for more MinAtar games.
>
> Studying actor-critic methods such as PPO or CQL would add a confounding factor since they simultaneously train a stochastic policy (as you correctly noted above). To isolate the effects of return estimation itself, we chose to focus on DQN, a well-studied deep RL method that only learns a value function.
>
> We also note that PPO is an on-policy method and commonly uses the $\lambda$-return with GAE [1] rather than the high-variance Monte Carlo return, which provides further evidence that variance reduction is beneficial in practice. Thus, the main benefit of PiLaR is to apply similar variance reduction in off-policy experience-replay settings where computing the full $\lambda$-return is infeasible.
>
> **References**
>
> [1] High-dimensional Continuous Control Using Generalized Advantage Estimation. John Schulman et al., ICLR 2016.

---

### Author Response · Authors · 2023-11-16
**General Response to All Reviewers**

We thank all the reviewers for their feedback. It seems to us that the recommendation for rejecting our paper stems from a misunderstanding about the significance (and consequence) of our theoretical results, and from the expectation of additional empirical validation of Piecewise $\lambda$-returns (PiLaRs). We respond to and address these general concerns below and we hope that we can iterate with the reviewers on these to minimize any anchoring effect based on the initial scores our paper received. We also respond to each individual concern in direct responses to the reviewers.

**Summary of Changes**

- Clarified strength of contributions in Introduction.
- Fixed minor typos in Eqs. (8) and (12).
- Added deep RL results for three more MinAtar games (Figures 3, 5, and 6).
- Added Section 3.2: *Finite-Time Analysis* (proof in Appendix B.2)
- Added Appendix A: *How Realistic Is the Proposed Variance Model?*, along with three new experiments to empirically validate our variance model.

The finite-time analysis is highlighted in **blue**. Other textual changes are highlighted in **red**.

**Strength of Theoretical Contributions**

There has been a misunderstanding regarding the significance and novelty of our theoretical results, possibly because we were not initially clear enough about the originality of our contributions. We have now added a couple of sentences in the new version of the paper to clarify this.

Our work is a major advance in the theory of compound returns. To the best of our knowledge, the last paper that developed new theoretical insights for compound returns is from 2011 [1]. Our results improve upon the existing theory in several ways:
1. **We are the first to explicitly model the variance of compound returns** (Proposition 2 and Theorem 1), which are arbitrary weighted averages of $n$-step returns. An $n$-step return is the simplest multistep return estimator, which bootstraps from the value function exactly $n$ time steps after the experience being reinforced. **Previous work derived a variance model only for these simple $n$-step returns** and then used it to justify a compound return called $\text{TD}_\gamma$. We are therefore the first to provide strong mathematical evidence for the benefit of all compound returns compared to $n$-step returns. This is a very general result and applies to virtually all existing return estimators since they can be expressed as compound returns.
2. Theorem 1 is a major contribution to the theory of compound returns. **Previously, no one had ever proved that compound returns are able to reduce variance relative to $n$-step returns without increasing bias.** The proof technique itself is also novel and nontrivial, as it involves deriving a general expression for the compound variance (see the previous point) and then re-expressing it in terms of the return’s contraction rate and deriving the final bound in terms of the $n$-step variance. We do not know of any papers that used this technique before.
3. **Although the uniform-correlation assumption on the TD errors is arguably strong, it is a significant relaxation from previous work [1] which assumed that all TD errors are uncorrelated**. Our work is the first to make progress on this question in the last 12 years. Importantly, notice that our random walk experiment (Figure 1) shows that our predictions are accurate: namely, the expected performance (bias) of the returns are equal but the variance of the compound return is lower than that of the n-step return, which in turn leads to better value-function error. This is the main point of experimental results in a theoretical paper such as ours—to test how the theory is reflected in practice, when the assumptions one is required to make do not always hold.

---

> ### Author Response · Authors · 2023-11-16
> **General Response to All Reviewers (continued)**
>
> **Why Variance Reduction Leads to Faster Learning**
>
> Some reviewers expressed doubt that reducing the variance of return estimates would lead to faster learning. To see why this is the case, value-based RL methods like DQN (which is based on Q-Learning) can be viewed as stochastic approximations to action-value iteration. For example, consider the proof of Proposition 4.5 in Neuro-Dynamic Programming [2, pp. 166-167]. All of the control methods in our work are equivalent to sample-based approximations of some operator $H \colon \mathbb{R}^m \to \mathbb{R}^m$ that transforms an action-value function $q \in \mathbb{R}^m$, where $m = {|\mathcal{S} \times \mathcal{A}|}$. By definition, the optimal action-value function $q_*$ is the unique fixed point of $H$. The greedy policy with respect to $q_*$ is guaranteed to be optimal, so converging to this fixed point faster means that the behavior policy improves faster on average, too. The methods considered in our paper are therefore equivalent to asynchronous versions of the following stochastic process:
>
> $q_{t+1} \gets (1-\alpha_t) q_t + \alpha_t (H q_t + \omega_t)$,
>
> where $\omega_t \in \mathbb{R}^m$ is zero-mean, finite-variance noise. If we remove the noise, we can set $\alpha_t = 1$ and converge to $q_*$ exponentially fast. Intuitively, each iterate $q_t$ remains in a shrinking hypercube whose side length decays by the contraction rate. With noise, the iterates no longer monotonically approach the fixed point, but annealing the step size $\alpha_t$ according to the Robbins-Monro conditions [3] does guarantee that they will eventually enter and then remain within each successively shrinking hypercube with probability 1. The number of iterations required to enter and stay inside a given hypercube depends on the magnitude of the conditional variance of $\omega_t$, which is the conditional variance of the return estimator in our case. It follows that if one return estimator has the same contraction rate but lower variance compared to another, it will require fewer samples on average to achieve the same proximity to $q_*$. This explains why $\lambda$-returns learn faster than $n$-step returns in our random walk experiment (Figure 1) when their contraction rates are equalized, since Theorem 1 guarantees the $\lambda$-returns have lower variance.
>
> **Additional Experiments**
>
> All of the reviewers recommended adding more experiments to the paper. We would like to remind the reviewers that our paper is theoretical in nature. Our theory is the main contribution and we believe it stands by itself given the depth of our analysis. We therefore have left a larger empirical study for future work. Our deep RL results are meant to be a proof of concept of the feasibility of the method and its applicability to nonlinear function approximation, where theoretical guarantees do not exist.
>
> Nevertheless, **we have added results for three additional MinAtar games (Breakout, Seaquest, and Space Invaders)** that further demonstrate the utility of PiLaRs compared to $n$-step returns. They further support the claims we make in the paper.
>
> **References**
>
> [1] TDγ: Re-evaluating Complex Backups in Temporal Difference Learning. Konidaris et al., NeurIPS 2011.
>
> [2] Neuro-Dynamic Programming. Bertsekas and Tsitsiklis, 1996.
>
> [3] A Stochastic Approximation Method. Robbins and Monro, 1951.

---

### Meta-Review · Area_Chair_jmG9 · 2023-12-13

**Metareview:**

This paper considers reinforcement learning and studies the variance of compound returns from both theoretical and empirical perspectives. The paper first proves that when experiences are not perfectly correlated, any compound return has lower variance than corresponding step return with the same contraction rate. The paper then propose an approximation of compound return in practice (PiLaR) and shows its effectiveness by running PiLaR on DQN for tabular examples. Reviewers raised many concerns towards the original submitted version of this paper, including theoretical results being a bit weak, the connection between proved results to improvement in performance is not clear, and experiments on rather limited settings. While we appreciate we authors' efforts in addressing several concerns in the rebuttal phase, it makes rather significant modifications to many parts of the paper, which may require another round of careful evaluation on the contribution conditioned on the new added results (all of which happen after the submission deadline). We therefore recommend rejection here and suggest authors to submit the new version to future venue.

**Justification For Why Not Higher Score:**

The theoretical and empirical results of the original paper is not strong enough for ICLR.

**Justification For Why Not Lower Score:**

N/A

---

### Decision · Program_Chairs · 2024-01-16

Reject